# Beta-CROWN: Efficient Bound Propagation with Per-neuron Split Constraints for Neural Network Robustness Verification

**Shiqi Wang**[*,1]    **Huan Zhang**[*,2]    **Kaidi Xu**[*,3]

**Xue Lin**[3]    **Suman Jana**[1]    **Cho-Jui Hsieh**[4]    **Zico Kolter**[2]

[1]Columbia University    [2]CMU    [3]Northeastern University    [4]UCLA

sw3215@columbia.edu   huan@huan-zhang.com   xu.kaid@northeastern.edu
xue.lin@northeastern.edu   suman@cs.columbia.edu   chohsieh@cs.ucla.edu
zkolter@cs.cmu.edu

*\* Equal Contribution*

## Abstract

Bound propagation based incomplete neural network verifiers such as CROWN are very efficient and can significantly accelerate branch-and-bound (BaB) based complete verification of neural networks. However, bound propagation cannot fully handle the neuron split constraints introduced by BaB commonly handled by expensive linear programming (LP) solvers, leading to loose bounds and hurting verification efficiency. In this work, we develop $\beta$-CROWN, a new bound propagation based method that can fully encode neuron splits via optimizable parameters $\boldsymbol{\beta}$ constructed from either primal or dual space. When jointly optimized in intermediate layers, $\beta$-CROWN generally produces better bounds than typical LP verifiers with neuron split constraints, while being as efficient and parallelizable as CROWN on GPUs. Applied to complete robustness verification benchmarks, $\beta$-CROWN with BaB is up to three orders of magnitude faster than LP-based BaB methods, and is notably faster than all existing approaches while producing lower timeout rates. By terminating BaB early, our method can also be used for efficient incomplete verification. We consistently achieve higher verified accuracy in many settings compared to powerful incomplete verifiers, including those based on convex barrier breaking techniques. Compared to the typically tightest but very costly semidefinite programming (SDP) based incomplete verifiers, we obtain higher verified accuracy with three orders of magnitudes less verification time. Our algorithm empowered the $\alpha,\beta$-CROWN (`alpha-beta-CROWN`) verifier, the winning tool in VNN-COMP 2021. Our code is available at http://PaperCode.cc/BetaCROWN.

## 1   Introduction

As neural networks (NNs) are being deployed in safety-critical applications, it becomes increasingly important to formally verify their behaviors under potentially malicious inputs. Broadly speaking, the neural network verification problem involves proving certain desired relationships between inputs and outputs (often referred to as *specifications*), such as safety or robustness guarantees, for all inputs inside some domain. Canonically, the problem can be cast as finding the global minima of some functions on the network's outputs (e.g., the difference between the predictions of the true label and another target label), within a bounded input set as constraints. This is a challenging problem due to the non-convexity and high dimensionality of neural networks.

35th Conference on Neural Information Processing Systems (NeurIPS 2021).

We first focus on *complete* verification: the verifier should give a definite "yes/no" answer given sufficient time. Many complete verifiers rely on the branch and bound (BaB) method [8] involving (1) branching by recursively splitting the original verification problem into subdomains (e.g., splitting a ReLU neuron into positive/negative linear regions by adding split constraints) and (2) bounding each subdomain with specialized incomplete verifiers. Traditional BaB-based verifiers use expensive linear programming (LP) solvers [15, 23, 7] as incomplete verifiers which can fully encode neuron split constraints. Meanwhile, a recent verifier, Fast-and-Complete [45], demonstrates that cheap incomplete verifiers can significantly accelerate complete verification on GPUs over LP-based ones thanks to their efficiency. Many cheap incomplete verifiers are based on *bound propagation methods* [46, 42, 41, 13, 17, 36, 44], i.e., maintaining and propagating tractable and sound bounds through networks, and CROWN [46] is a representative which propagates a linear or quadratic bound.

However, unlike LP based verifiers, existing bound propagation methods lack the power to handle neuron split constraints introduced by BaB. For instance, given inputs $x, y \in [-1, 1]$, they can bound a ReLU's input $x + y$ as $[-2, 2]$ but they have no means to consider neuron split constraints such as $x - y \geq 0$ introduced by splitting another ReLU to the positive linear region. Such a problem causes looser bounds and unnecessary branching, hurting the verification efficiency. Even worse, without considering these split constraints, bound propagation methods cannot detect many infeasible subdomains in BaB [45], leading to incompleteness unless costly checking is performed.

In our work, we develop a new, fast bound propagation based incomplete verifier, $\beta$-CROWN. It solves an optimization problem equivalent to the expensive LP based methods with neuron split constraints while still enjoying the efficiency of bound propagation methods. $\beta$-CROWN contains optimizable parameters $\boldsymbol{\beta}$ which come from propagation of Lagrangian multipliers, and any valid settings of these parameters yield sound bounds for verification. These parameters are optimized using a few steps of (super)gradient ascent to achieve bounds as tight as possible. Optimizing $\boldsymbol{\beta}$ can also eliminate many infeasible subdomains and avoid further useless branching. Furthermore, we can jointly optimize intermediate layer bounds similar to [44] but also with the additional parameters $\boldsymbol{\beta}$, allowing $\beta$-CROWN to tighten relaxations and outperform typical LP verifiers with fixed intermediate layer bounds. Unlike traditional LP-based BaB methods, $\beta$-CROWN can be efficiently implemented with an automatic differentiation framework on GPUs to fully exploit the power of modern accelerators. The combination of $\beta$-CROWN and BaB ($\beta$-CROWN BaB) produces a complete verifier with GPU acceleration, reducing the verification time of traditional LP based BaB verifiers [8] by up to *three orders of magnitudes* on a commonly used benchmark suite on CIFAR-10 [6, 10]. Compared to all state-of-the-art GPU-based complete verifiers [7, 45, 10, 23, 6, 11], our approach is noticeably faster with lower timeout rates. Our algorithm empowered the tool $\alpha,\beta$-CROWN (`alpha-beta-CROWN`), which won the 2nd International Verification of Neural Networks Competition [3] (VNN-COMP 2021) with the highest total score and verified the most number of problem instances in 8 benchmarks.

Finally, by terminating our complete verifier $\beta$-CROWN BaB early, our approach can also function as a more accurate incomplete verifier by returning an incomplete but sound lower bound of all subdomains explored so far. We achieve better verified accuracy on a few benchmarking models over powerful incomplete verifiers including those based on tight linear relaxations [35, 37, 26] and semidefinite relaxations [9]. Compared to the typically tightest but very costly incomplete verifier SDP-FO [9] based on the semidefinite programming (SDP) relaxations [28, 14], our method obtains consistently higher verified accuracy while reducing verification time by three orders of magnitudes.

## 2 Background

### 2.1 The neural network verification problem and its LP relaxation

We define the input of a neural network as $x \in \mathbb{R}^{d_0}$, and define the weights and biases of an $L$-layer neural network as $\mathbf{W}^{(i)} \in \mathbb{R}^{d_i \times d_{i-1}}$ and $\mathbf{b}^{(i)} \in \mathbb{R}^{d_i}$ ($i \in \{1, \cdots, L\}$) respectively. For simplicity we assume that $d_L = 1$ so $\mathbf{W}^{(L)}$ is a vector and $\mathbf{b}^{(L)}$ is a scalar. The neural network function $f : \mathbb{R}^{d_0} \to \mathbb{R}$ is defined as $f(x) = z^{(L)}(x)$, where $z^{(i)}(x) = \mathbf{W}^{(i)} \hat{z}^{(i-1)}(x) + \mathbf{b}^{(i)}$, $\hat{z}^{(i)}(x) = \sigma(z^{(i)}(x))$ and $\hat{z}^{(0)}(x) = x$. $\sigma$ is the activation function and we use ReLU throughout this paper. When the context is clear, we omit $\cdot(x)$ and use $z_j^{(i)}$ and $\hat{z}_j^{(i)}$ to represent the *pre-activation* and *post-activation* values of the $j$-th neuron in the $i$-th layer. Neural network verification seeks the solution of the optimization problem in Eq. 1:

$$\min f(x) := z^{(L)}(x) \quad \text{s.t.} \ z^{(i)} = \mathbf{W}^{(i)} \hat{z}^{(i-1)} + \mathbf{b}^{(i)}, \hat{z}^{(i)} = \sigma(z^{(i)}), x \in \mathcal{C}, i \in \{1, \cdots, L-1\} \quad (1)$$

The set $\mathcal{C}$ defines the allowed input region and our aim is to find the minimum of $f(x)$ for $x \in \mathcal{C}$, and throughout this paper we consider $\mathcal{C}$ as an $\ell_\infty$ ball around a data example $x_0$: $\mathcal{C} = \{x \mid \|x - x_0\|_\infty \leq \epsilon\}$ but other $\ell_p$ norms can also be supported. In practical settings, we typically have "specifications" to verify, which are (usually linear) functions of neural network outputs describing the desired behavior of neural networks. For example, to guarantee robustness we typically investigate the margin between logits. Because the specification can also be seen as an output layer of NN and merged into $f(x)$ under verification, we do not discuss it in detail in this work. We consider the canonical specification $f(x) > 0$: if we can prove that $f(x) > 0$, $\forall x \in \mathcal{C}$, we say $f(x)$ is verified.

When $\mathcal{C}$ is a convex set, Eq. 1 is still a non-convex problem because the constraints $\hat{z}^{(i)} = \sigma(z^{(i)})$ are non-convex. Given unlimited time, *complete* verifiers can solve Eq. 1 exactly: $f^* = \min f(x)$, $\forall x \in \mathcal{C}$, so we can always conclude if the specification holds or not for any problem instance. On the other hand, *incomplete* verifiers usually relax the non-convexity of neural networks to obtain a tractable lower bound of the solution $\underline{f} \leq f^*$. If $\underline{f} \geq 0$, then $f^* > 0$ so $f(x)$ can be verified; when $\underline{f} < 0$, we are not able to infer the sign of $f^*$ so cannot conclude if the specification holds or not.

A commonly used incomplete verification technique is to relax non-convex ReLU constraints with linear constraints and turn the verification problem into a linear programming (LP) problem, which can then be solved with linear solvers. We refer to it as the "LP verifier" in this paper. Specifically, given $\text{ReLU}(z_j^{(i)}) := \max(0, z_j^{(i)})$ and its intermediate layer bounds $\mathbf{l}_j^{(i)} \leq z_j^{(i)} \leq \mathbf{u}_j^{(i)}$, each ReLU can be categorized into three cases: (1) if $\mathbf{l}_j^{(i)} \geq 0$ (ReLU in linear region) then $\hat{z}_j^{(i)} = z_j^{(i)}$; (2) if $\mathbf{u}_j^{(i)} \leq 0$ (ReLU in inactive region) then $\hat{z}_j^{(i)} = 0$; (3) if $\mathbf{l}_j^{(i)} \leq 0 \leq \mathbf{u}_j^{(i)}$ (ReLU is *unstable*) then three linear bounds are used: $\hat{z}_j^{(i)} \geq 0$, $\hat{z}_j^{(i)} \geq z_j^{(i)}$, and $\hat{z}_j^{(i)} \leq \frac{\mathbf{u}_j^{(i)}}{\mathbf{u}_j^{(i)} - \mathbf{l}_j^{(i)}}\left(z_j^{(i)} - \mathbf{l}_j^{(i)}\right)$; they are often referred to as the "triangle" relaxation [15, 42]. The intermediate layer bounds $\mathbf{l}^{(i)}$ and $\mathbf{u}^{(i)}$ are usually obtained from a cheaper bound propagation method (see next subsection). LP verifiers can provide relatively tight bounds but linear solvers are still expensive especially when the network is large. Also, unlike our $\beta$-CROWN, they have to use fixed intermediate bounds and cannot use the joint optimization of intermediate layer bounds (Section 3.3) to tighten relaxation.

## 2.2 CROWN: efficient incomplete verification by propagating linear bounds

Another cheaper way to give a lower bound for the objective in Eq. 1 is through sound bound propagation. CROWN [46] is a representative method that propagates a linear bound of $f(x)$ w.r.t. every intermediate layer in a backward manner until reaching the input $x$. CROWN uses two linear constraints to relax unstable ReLU neurons: a linear upper bound $\hat{z}_j^{(i)} \leq \frac{\mathbf{u}_j^{(i)}}{\mathbf{u}_j^{(i)} - \mathbf{l}_j^{(i)}}\left(z_j^{(i)} - \mathbf{l}_j^{(i)}\right)$ and a linear lower bound $\hat{z}_j^{(i)} \geq \boldsymbol{\alpha}_j^{(i)} z_j^{(i)}$ $(0 \leq \boldsymbol{\alpha}_j^{(i)} \leq 1)$. We can then bound the output of a ReLU layer:

**Lemma 2.1** (ReLU relaxation in CROWN). *Given $w, v \in \mathbb{R}^d$, $\mathbf{l} \leq v \leq \mathbf{u}$ (element-wise), we have*
$$w^\top \text{ReLU}(v) \geq w^\top \mathbf{D} v + b',$$
*where $\mathbf{D}$ is a diagonal matrix containing free variables $0 \leq \boldsymbol{\alpha}_j \leq 1$ only when $\mathbf{u}_j > 0 > \mathbf{l}_j$ and $w_j \geq 0$, while its rest values as well as constant $b'$ are determined by $\mathbf{l}, \mathbf{u}, w$.*

Detailed forms of each term are listed in Appendix A. Lemma 2.1 can be repeatedly applied, resulting in an efficient back-substitution procedure to derive a linear lower bound of NN output w.r.t. $x$:

**Lemma 2.2** (CROWN bound [46]). *Given an $L$-layer ReLU NN $f(x) : \mathbb{R}^{d_0} \to \mathbb{R}$ with weights $\mathbf{W}^{(i)}$, biases $\mathbf{b}^{(i)}$, pre-ReLU bounds $\mathbf{l}^{(i)} \leq z^{(i)} \leq \mathbf{u}^{(i)}$ ($1 \leq i \leq L$) and input constraint $x \in \mathcal{C}$. We have*
$$\min_{x \in \mathcal{C}} f(x) \geq \min_{x \in \mathcal{C}} \boldsymbol{a}_{\text{CROWN}}^\top x + c_{\text{CROWN}}$$
*where $\boldsymbol{a}_{\text{CROWN}}$ and $c_{\text{CROWN}}$ can be computed using $\mathbf{W}^{(i)}, \mathbf{b}^{(i)}, \mathbf{l}^{(i)}, \mathbf{u}^{(i)}$ in polynomial time.*

When $\mathcal{C}$ is an $\ell_p$ norm ball, minimization over the linear function can be easily solved using Hölder's inequality. The main benefit of CROWN is its efficiency: CROWN can be efficiently implemented on machine learning accelerators such as GPUs [44] and TPUs [47], and it can be a few magnitudes faster than an LP verifier which is hard to parallelize on GPUs. CROWN was generalized to general architectures [44, 31] while we only demonstrate it for feedforward ReLU networks for simplicity. Additionally, Xu et al. [45] showed that it is possible to optimize the slope of the lower bound, $\boldsymbol{\alpha}$, using gradient ascent, to further tighten the bound (sometimes referred to as $\alpha$-CROWN).

## 2.3 Branch and Bound and Neuron Split Constraints

Branch and bound (BaB) method is widely adopted in complete verifiers [8]: we divide the domain of the verification problem $\mathcal{C}$ into two subdomains $\mathcal{C}_1 = \{x \in \mathcal{C}, z_j^{(i)} \geq 0\}$ and $\mathcal{C}_2 = \{x \in \mathcal{C}, z_j^{(i)} < 0\}$ where $z_j^{(i)}$ is an unstable ReLU neuron in $\mathcal{C}$ but now becomes linear for each subdomain. Incomplete verifiers can then estimate the lower bound of each subdomain with relaxations. If the lower bound produced for subdomain $\mathcal{C}_i$ (denoted by $\underline{f}_{\mathcal{C}_i}$) is greater than 0, $\mathcal{C}_i$ is verified; otherwise, we further branch over domain $\mathcal{C}_i$ by splitting another unstable ReLU neuron. The process terminates when all subdomains are verified. The completeness is guaranteed when all unstable ReLU neurons are split.

**LP verifier with neuron split constraints.** A popular incomplete verifier used in BaB is the LP verifier. Essentially, when we split the $j$-th ReLU in layer $i$, we can simply add $z_j^{(i)} \geq 0$ or $z_j^{(i)} < 0$ to Eq. 1 and get a linearly relaxed lower bound to each subdomain. We denote the $\mathcal{Z}^{+(i)}$ and $\mathcal{Z}^{-(i)}$ as the set of neuron indices with positive and negative split constraints in layer $i$. We define the split constraints at layer $i$ as $\mathcal{Z}^{(i)} := \{z^{(i)} \mid z_{j_1}^{(i)} \geq 0, z_{j_2}^{(i)} < 0, \forall j_1 \in \mathcal{Z}^{+(i)}, \forall j_2 \in \mathcal{Z}^{-(i)}\}$. We denote the vector of all pre-ReLU neurons as $z$, and we define a set $\mathcal{Z}$ to represent the split constraints on $z$: $\mathcal{Z} = \mathcal{Z}^{(1)} \cap \mathcal{Z}^{(2)} \cap \cdots \cap \mathcal{Z}^{(L-1)}$. For convenience, we also use the shorthand $\tilde{\mathcal{Z}}^{(i)} := \mathcal{Z}^{(1)} \cap \cdots \cap \mathcal{Z}^{(i)}$ and $\tilde{z}^{(i)} := \{z^{(1)}, z^{(2)}, \cdots, z^{(i)}\}$. LP verifiers can easily handle these neuron split constraints but are more expensive than bound propagation methods like CROWN and cannot be accelerated on GPUs.

**Branching strategy.** Branching strategies (selecting which ReLU neuron to split) are generally agnostic to the incomplete verifier used in BaB but do affect the overall BaB performance. BaBSR [7] is a widely used strategy in complete verifiers, which is based on an fast estimates on objective improvements after splitting each neuron. The neuron with highest estimated improvement is selected for branching. Recently, Filtered Smart Branching (FSB) [11] improves BaBSR by mimicking strong branching - it utilizes bound propagation methods to evaluate the best a few candidates proposed by BaBSR and chooses the one with largest improvement. Graph neural network (GNN) based branching was also proposed [23]. Our $\beta$-CROWN BaB is a general complete verification framework fit for any potential branching strategy, and we evaluate both BaBSR and FSB in experiments.

## 3 $\beta$-CROWN for Complete and Incomplete Verification

In this section, we first give intuitions on how $\beta$-CROWN handles neuron split constraints without costly LP solvers. Then we formally state the main theorem of $\beta$-CROWN from both primal and dual spaces, and discuss how to tighten the bounds using free parameters $\boldsymbol{\alpha}, \boldsymbol{\beta}$. Lastly, we propose $\beta$-CROWN BaB, a complete verifier that is also a strong incomplete verifier when stopped early.

### 3.1 $\beta$-CROWN: Linear Bound Propagation with Neuron Split Constraints

The NN verification problem under neuron split constraints can be seen as an optimization problem:

$$\min_{x \in \mathcal{C}, z \in \mathcal{Z}} f(x). \tag{2}$$

Bound propagation methods like CROWN can give a relatively tight lower bound for $\min_{x \in \mathcal{C}} f(x)$ but they *cannot handle the neuron split constraints $z \in \mathcal{Z}$*. Before we present our main theorem, we first show the intuition on how to apply split constraints to the bound propagation process.

To encode the neuron splits, we first define diagonal matrix $\mathbf{S}^{(i)} \in \mathbb{R}^{d_i \times d_i}$ in Eq. 3 where $i \in [1, \cdots L-1], j \in [1, \cdots, d_i]$ are indices of layers and neurons, respectively:

$$\mathbf{S}_{j,j}^{(i)} = -1(\text{if split } z_j^{(i)} \geq 0); \quad \mathbf{S}_{j,j}^{(i)} = +1(\text{if split } z_j^{(i)} < 0); \quad \mathbf{S}_{j,j}^{(i)} = 0(\text{if no split } z_j^{(i)}) \tag{3}$$

We start from the last layer and derive linear bounds for each intermediate layer $z^{(i)}$ and $\hat{z}^{(i)}$ with both constraints $x \in \mathcal{C}$ and $z \in \mathcal{Z}$. We also assume that pre-ReLU bounds $\mathbf{l}^{(i)} \leq z^{(i)} \leq \mathbf{u}^{(i)}$ for each layer $i$ are available (see discussions in Sec. 3.3 on these intermediate layer bounds). We initially have:

$$\min_{x \in \mathcal{C}, z \in \mathcal{Z}} f(x) = \min_{x \in \mathcal{C}, z \in \mathcal{Z}} \mathbf{W}^{(L)} \hat{z}^{(L-1)} + \mathbf{b}^{(L)}. \tag{4}$$

Since $\hat{z}^{(L-1)} = \text{ReLU}(z^{(L-1)})$, we can apply Lemma 2.1 to relax the ReLU neuron at layer $L-1$, and obtain a linear lower bound for $f(x)$ w.r.t. $z^{(L-1)}$ (we omit all constant terms to avoid clutter):

$$\min_{x\in\mathcal{C},z\in\mathcal{Z}} f(x) \geq \min_{x\in\mathcal{C},z\in\mathcal{Z}} \mathbf{W}^{(L)}\mathbf{D}^{(L-1)}z^{(L-1)} + \text{const.}$$

To enforce the split neurons at layer $L-1$, we use a Lagrange function with $\boldsymbol{\beta}^{(L-1)\top}\mathbf{S}^{(L-1)}$ multiplied on $z^{(L-1)}$:

$$
\begin{aligned}
\min_{x\in\mathcal{C},z\in\mathcal{Z}} f(x) &\geq \min_{\substack{x\in\mathcal{C}\\ \tilde{z}^{(L-2)}\in\tilde{\mathcal{Z}}^{(L-2)}}} \max_{\boldsymbol{\beta}^{(L-1)}\geq 0} \mathbf{W}^{(L)}\mathbf{D}^{(L-1)}z^{(L-1)} + \boldsymbol{\beta}^{(L-1)^\top}\mathbf{S}^{(L-1)}z^{(L-1)} + \text{const}\\
&\geq \max_{\boldsymbol{\beta}^{(L-1)}\geq 0} \min_{\substack{x\in\mathcal{C}\\ \tilde{z}^{(L-2)}\in\tilde{\mathcal{Z}}^{(L-2)}}} \left(\mathbf{W}^{(L)}\mathbf{D}^{(L-1)} + \boldsymbol{\beta}^{(L-1)^\top}\mathbf{S}^{(L-1)}\right)z^{(L-1)} + \text{const}
\end{aligned}
\tag{5}
$$

The first inequality is due to the definition of the Lagrange function: we remove the constraint $z^{(L-1)} \in \mathcal{Z}^{(L-1)}$ and use a multiplier to replace this constraint. The second inequality is due to weak duality. Due to the design of $\mathbf{S}^{(L-1)}$, neuron split $z_j^{(L-1)} \geq 0$ has a negative multiplier $-\boldsymbol{\beta}_j^{(L-1)}$ and split $z_j^{(L-1)} < 0$ has a positive multiplier $\boldsymbol{\beta}_j^{(L-1)}$. Any $\boldsymbol{\beta}^{(L-1)} \geq 0$ yields a lower bound for the constrained optimization problem. Then we substitute $z^{(L-1)}$ with $\mathbf{W}^{(L-1)}\hat{z}^{(L-2)} + \mathbf{b}^{(L-1)}$ for next layer:

$$\min_{x\in\mathcal{C},z\in\mathcal{Z}} f(x) \geq \max_{\boldsymbol{\beta}^{(L-1)}\geq 0} \min_{\substack{x\in\mathcal{C}\\ \tilde{z}^{(L-2)}\in\tilde{\mathcal{Z}}^{(L-2)}}} \left(\mathbf{W}^{(L)}\mathbf{D}^{(L-1)} + \boldsymbol{\beta}^{(L-1)^\top}\mathbf{S}^{(L-1)}\right)\mathbf{W}^{(L-1)}\hat{z}^{(L-2)} + \text{const} \tag{6}$$

We define a matrix $\mathbf{A}^{(i)}$ to represent the linear relationship between $f(x)$ and $\hat{z}^{(i)}$, where $\mathbf{A}^{(L-1)} = \mathbf{W}^{(L)}$ according to Eq. 4 and $\mathbf{A}^{(L-2)} = (\mathbf{A}^{(L-1)}\mathbf{D}^{(L-1)} + \boldsymbol{\beta}^{(L-1)^\top}\mathbf{S}^{(L-1)})\mathbf{W}^{(L-1)}$ by Eq. 6. Considering 1-dimension output $f(x)$, $\mathbf{A}^{(i)}$ has only 1 row. With $\mathbf{A}^{(L-2)}$, Eq. 6 becomes:

$$\min_{x\in\mathcal{C},z\in\mathcal{Z}} f(x) \geq \max_{\boldsymbol{\beta}^{(L-1)}\geq 0} \min_{\substack{x\in\mathcal{C}\\ \tilde{z}^{(L-2)}\in\tilde{\mathcal{Z}}^{(L-2)}}} \mathbf{A}^{(L-2)}\hat{z}^{(L-2)} + \text{const},$$

which is in a form similar to Eq. 4 except for the outer maximization over $\boldsymbol{\beta}^{(L-1)}$. This allows the back-substitution process (Eq. 4, Eq. 5, and Eq. 6) to continue. In each step, we swap $\max$ and $\min$ as in Eq. 5, so every maximization over $\boldsymbol{\beta}^{(i)}$ is outside of $\min_{x\in\mathcal{C}}$. Eventually, we have:

$$\min_{x\in\mathcal{C},z\in\mathcal{Z}} f(x) \geq \max_{\boldsymbol{\beta}\geq 0} \min_{x\in\mathcal{C}} \mathbf{A}^{(0)}x + \text{const},$$

where $\boldsymbol{\beta} := \begin{bmatrix} \boldsymbol{\beta}^{(1)\top} & \boldsymbol{\beta}^{(2)\top} & \cdots & \boldsymbol{\beta}^{(L-1)\top} \end{bmatrix}^\top$ concatenates all $\boldsymbol{\beta}^{(i)}$ vectors. Following the above idea, we present the main theorem in Theorem 3.1 (proof is given in Appendix A).

**Theorem 3.1** ($\beta$-CROWN bound). *Given an $L$-layer NN $f(x) : \mathbb{R}^{d_0} \to \mathbb{R}$ with weights $\mathbf{W}^{(i)}$, biases $\mathbf{b}^{(i)}$, pre-ReLU bounds $\mathbf{l}^{(i)} \leq z^{(i)} \leq \mathbf{u}^{(i)}$ ($1 \leq i \leq L$), input bounds $\mathcal{C}$, split constraints $\mathcal{Z}$. We have:*

$$\min_{x\in\mathcal{C},z\in\mathcal{Z}} f(x) \geq \max_{\boldsymbol{\beta}\geq 0} \min_{x\in\mathcal{C}} (\boldsymbol{a} + \mathbf{P}\boldsymbol{\beta})^\top x + \mathbf{q}^\top\boldsymbol{\beta} + c, \tag{7}$$

*where $\boldsymbol{a} \in \mathbb{R}^{d_0}, \mathbf{P} \in \mathbb{R}^{d_0\times(\sum_{i=1}^{L-1}d_i)}, \mathbf{q} \in \mathbb{R}^{\sum_{i=1}^{L-1}d_i}$ and $c \in \mathbb{R}$ are functions of $\mathbf{W}^{(i)}$, $\mathbf{b}^{(i)}$, $\mathbf{l}^{(i)}$, $\mathbf{u}^{(i)}$.*

Detailed formulations for $\boldsymbol{a}$, $\mathbf{P}$, $\mathbf{q}$ and $c$ are given in Appendix A. Theorem 3.1 shows that when neuron split constraints exist, $f(x)$ can still be bounded by a linear equation containing optimizable multipliers $\boldsymbol{\beta}$. Observing Eq. 5, the main difference between CROWN and $\beta$-CROWN lies in the relaxation of each ReLU layer, where we need an extra term $\boldsymbol{\beta}^{(i)\top}\mathbf{S}^{(i)}$ in the linear relationship matrix (for example, $\mathbf{W}^{(L)}\mathbf{D}^{(L-1)}$ in Eq. 5) between $f(x)$ and $z^{(i)}$ to enforce neuron split constraints. This extra term in every ReLU layer yields $\mathbf{P}$ and $\mathbf{q}$ in Eq. 7 after bound propagations.

To solve the optimization problem in Eq. 7, we note that in the $\ell_p$ norm robustness setting ($\mathcal{C} = \{x \mid \|x - x_0\|_p \leq \epsilon\}$), the inner minimization has a closed solution:

$$\min_{x\in\mathcal{C},z\in\mathcal{Z}} f(x) \geq \max_{\boldsymbol{\beta}\geq 0} -\|\boldsymbol{a} + \mathbf{P}\boldsymbol{\beta}\|_q\epsilon + (\mathbf{P}^\top x_0 + \mathbf{q})^\top\boldsymbol{\beta} + \boldsymbol{a}^\top x_0 + c := \max_{\boldsymbol{\beta}\geq 0} g(\boldsymbol{\beta}) \tag{8}$$

where $\frac{1}{p} + \frac{1}{q} = 1$. The maximization is concave in $\boldsymbol{\beta}$ ($q \geq 1$), so we can simply optimize it using projected (super)gradient ascent with gradients from an automatic differentiation library. Since any

$\boldsymbol{\beta} \geq 0$ yields a valid lower bound for $\min_{x \in \mathcal{C}, z \in \mathcal{Z}} f(x)$, convergence is not necessary to guarantee soundness. $\beta$-CROWN is efficient - it has the same asymptotic complexity as CROWN when $\boldsymbol{\beta}$ is fixed. When $\boldsymbol{\beta} = 0$, $\beta$-CROWN yields the same results as CROWN; however the additional optimizable $\boldsymbol{\beta}$ allows us to maximize and tighten the lower bound due to neuron split constraints.

We define $\boldsymbol{\alpha}^{(i)} \in \mathbb{R}^{d_i}$ for free variables associated with unstable ReLU neurons in Lemma 2.1 for layer $i$ and define all free variables $\boldsymbol{\alpha} = \{\boldsymbol{\alpha}^{(1)} \cdots \boldsymbol{\alpha}^{(L-1)}\}$. Since any $0 \leq \boldsymbol{\alpha}_j^{(i)} \leq 1$ yields a valid bound, we can optimize it to tighten the bound, similarly as done in [45]. Formally, we rewrite Eq. 8 with $\boldsymbol{\alpha}$ explicitly:

$$\min_{x \in \mathcal{C}, z \in \mathcal{Z}} f(x) \geq \max_{0 \leq \boldsymbol{\alpha} \leq 1, \boldsymbol{\beta} \geq 0} g(\boldsymbol{\alpha}, \boldsymbol{\beta}). \tag{9}$$

## 3.2 Connections to the Dual Problem

In this subsection, we show that $\beta$-CROWN can also be derived from a dual LP problem. Based on Eq. 1 and linear relaxations in Section 2, we first construct an LP problem for $\ell_\infty$ robustness verification in Eq. 10 where $i \in \{1, \cdots, L-1\}$.

$$\min f(x) := z^{(L)}(x) \quad \text{s.t.}$$

Network and Input Bounds: $z^{(i)} = \mathbf{W}^{(i)} \hat{z}^{(i-1)} + \mathbf{b}^{(i)}; \hat{z}^{(0)} \geq x_0 - \epsilon; \hat{z}^{(0)} \leq x_0 + \epsilon;$

Stable ReLUs: $\hat{z}_j^{(i)} = z_j^{(i)}$ (if $\mathbf{l}_j^{(i)} \geq 0$); $\hat{z}_j^{(i)} = 0$ (if $\mathbf{u}_j^{(i)} \leq 0$);

Unstable: $\hat{z}_j^{(i)} \geq 0, \hat{z}_j^{(i)} \geq z_j^{(i)}, \hat{z}_j^{(i)} \leq \frac{\mathbf{u}_j^{(i)}}{\mathbf{u}_j^{(i)} - \mathbf{l}_j^{(i)}} \left( z_j^{(i)} - \mathbf{l}_j^{(i)} \right)$ (if $\mathbf{l}_j^{(i)} < 0 < \mathbf{u}_j^{(i)}, j \notin \mathcal{Z}^{+(i)} \cup \mathcal{Z}^{-(i)}$)

Neuron Split Constraints: $\hat{z}_j^{(i)} = z_j^{(i)}, z_j^{(i)} \geq 0$ (if $j \in \mathcal{Z}^{+(i)}$); $\hat{z}_j^{(i)} = 0, z_j^{(i)} < 0$ (if $j \in \mathcal{Z}^{-(i)}$)
$$\tag{10}$$

Compared to the formulation in [42], we have neuron split constraints. Many BaB based complete verifiers [8, 23] use an LP solver for Eq. 10 as the incomplete verifier. We first show that it is possible to derive Theorem 3.1 from the dual of this LP, leading to Theorem 3.2:

**Theorem 3.2.** *The objective $d_{LP}$ for the dual problem of Eq. 10 can be represented as*

$$d_{LP} = -\|\boldsymbol{a} + \mathbf{P}\boldsymbol{\beta}\|_1 \cdot \epsilon + (\mathbf{P}^\top x_0 + \mathbf{q})^\top \boldsymbol{\beta} + \boldsymbol{a}^\top x_0 + c,$$

*where $\boldsymbol{a}$, $\mathbf{P}$, $\mathbf{q}$ and $c$ are defined in the same way as in Theorem 3.1, and $\boldsymbol{\beta} \geq 0$ corresponds to the dual variables of neuron split constraints in Eq. 10.*

A similar connection between CROWN and dual LP based verifier [42] was shown in [30], and their results can be seen as a special case of ours when $\boldsymbol{\beta} = 0$ (none of the split constraints are active). An immediate consequence is that $\beta$-CROWN can potentially solve Eq. 10 as well as using an LP solver:

**Corollary 3.2.1.** *When $\boldsymbol{\alpha}$ and $\boldsymbol{\beta}$ are optimally set and intermediate bounds $\mathbf{l}$, $\mathbf{u}$ are fixed, $\beta$-CROWN produces $p_{LP}^*$, the optimal objective of LP with split constraints in Eq. 10:*

$$\max_{0 \leq \boldsymbol{\alpha} \leq 1, \boldsymbol{\beta} \geq 0} g(\boldsymbol{\alpha}, \boldsymbol{\beta}) = p_{LP}^*,$$

In Appendix A, we give detailed formulations for conversions between the variables $\boldsymbol{\alpha}$, $\boldsymbol{\beta}$ in $\beta$-CROWN and their corresponding dual variables in the LP problem.

## 3.3 Joint Optimization of Free Variables in $\beta$-CROWN

In Eq. 9, $g$ is also a function of $\mathbf{l}_j^{(i)}$ and $\mathbf{u}_j^{(i)}$, the intermediate layer bounds for each neuron $z_j^{(i)}$. They are also computed using $\beta$-CROWN. To obtain $\mathbf{l}_j^{(i)}$, we set $f(x) := z_j^{(i)}(x)$ and apply Theorem 3.1:

$$\min_{x \in \mathcal{C}, \tilde{z}^{(i-1)} \in \tilde{\mathcal{Z}}^{(i-1)}} z_j^{(i)}(x) \geq \max_{0 \leq \boldsymbol{\alpha}' \leq 1, \boldsymbol{\beta}' \geq 0} g'(\boldsymbol{\alpha}', \boldsymbol{\beta}') := \mathbf{l}_j^{(i)} \tag{11}$$

For computing $\mathbf{u}_j^{(i)}$ we simply set $f(x) := -z_j^{(i)}(x)$. Importantly, during solving these intermediate layer bounds, the $\boldsymbol{\alpha}'$ and $\boldsymbol{\beta}'$ are *independent sets of variables*, not the same ones for the objective $f(x) := z^{(L)}$. Since $g$ is a function of $\mathbf{l}_j^{(i)}$, it is also a function of $\boldsymbol{\alpha}'$ and $\boldsymbol{\beta}'$. In fact, there are a total

of $\sum_{i=1}^{L-1} d_i$ intermediate layer neurons, and each neuron is associated with a set of independent $\boldsymbol{\alpha}'$ and $\boldsymbol{\beta}'$ variables. Optimizing these variables allowing us to tighten the relaxations on unstable ReLU neurons (which depend on $\mathbf{l}_j^{(i)}$ and $\mathbf{u}_j^{(i)}$) and produce tight final bounds, which is impossible in LP. In other words, we need to optimize $\hat{\boldsymbol{\alpha}}$ and $\hat{\boldsymbol{\beta}}$, which are two vectors concatenating $\boldsymbol{\alpha}, \boldsymbol{\beta}$ as well as a large number of $\boldsymbol{\alpha}'$ and $\boldsymbol{\beta}'$ used to compute each intermediate layer bound:

$$\min_{x \in \mathcal{C}, z \in \mathcal{Z}} f(x) \geq \max_{0 \leq \hat{\boldsymbol{\alpha}} \leq 1, \, \hat{\boldsymbol{\beta}} \geq 0} g(\hat{\boldsymbol{\alpha}}, \hat{\boldsymbol{\beta}}). \tag{12}$$

This formulation is non-convex and has a large number of variables. Since any $0 \leq \hat{\boldsymbol{\alpha}} \leq 1, \hat{\boldsymbol{\beta}} \geq 0$ leads to a valid lower bound, the non-convexity does not affect soundness. When intermediate layer bounds are also allowed to be tightened during optimization, we can outperform the LP verifier for Eq. 10 using fixed intermediate layer bounds. Typically, in many previous works [8, 23, 6], when the LP formulation Eq. 10 is formed, intermediate layer bounds are pre-computed with bound propagation procedures [8, 23], which are far from optimal. To estimate the dimension of this problem, we denote the number of unstable neurons at layer $i$ as $s_i := \mathrm{Tr}(|\mathbf{S}^{(i)}|)$. Each neuron in layer $i$ is associated with $2 \times \sum_{k=1}^{i-1} s_k$ variables $\boldsymbol{\alpha}'$. Suppose each hidden layer has $d$ neurons ($s_i = O(d)$), then $\hat{\boldsymbol{\alpha}}$ has $2 \times \sum_{i=1}^{L-1} d_i \sum_{k=1}^{i-1} s_k = O(L^2 d^2)$ variables in total. This can be too large for efficient optimization, so we share $\boldsymbol{\alpha}'$ and $\boldsymbol{\beta}'$ among the intermediate neurons of the same layer, leading to a total number of $O(L^2 d)$ variables to optimize. Note that a weaker form of joint optimization was also discussed in [45] without $\boldsymbol{\beta}$, and a detailed analysis can be found in Appendix B.2.

### 3.4 $\beta$-CROWN with Branch and Bound ($\beta$-CROWN BaB)

We perform complete verification following BaB framework [8] using $\beta$-CROWN as the incomplete solver, and we use simple branching heuristics like BaBSR [7] or FSB [11]. To efficiently utilize GPU, we also use batch splits to evaluate multiple subdomains in the same batch as in [44, 10]. We list our full algorithm $\beta$-CROWN BaB in Appendix B and we show it is sound and complete here:

**Theorem 3.3.** *$\beta$-CROWN with Branch and Bound on splitting ReLUs is sound and complete.*

Soundness is trivial because $\beta$-CROWN is a sound verifier. For completeness, it suffices to show that when all unstable ReLU neurons are split, $\beta$-CROWN gives the global minimum for Eq. 10. In contrast, combining CROWN [46] with BaB does *not* yield a complete verifier, as it cannot detect infeasible splits and a slow LP solver is still needed to guarantee completeness [45]. Instead, $\beta$-CROWN can detect infeasible subdomains - according to duality theory, an infeasible primal problem leads to an unbounded dual objective, which can be detected (see Sec. B.3 for more details).

Additionally, we show the potential of *early stopping a complete verifier as an incomplete verifier*. BaB approaches the exact solution of Eq. 1 by splitting the problem into multiple subdomains, and more subdomains give a tighter lower bound for Eq. 1. Unlike traditional complete verifiers, $\beta$-CROWN is efficient to explore a large number of subdomains during a very short time, making $\beta$-CROWN BaB an attractive solution for efficient incomplete verification.

## 4 Experimental Results

### 4.1 Comparison to Complete Verifiers

We evaluate complete verification performance on dataset provided in [23, 10] and used in VNN-COMP 2020 [22]. The benchmark contains three CIFAR-10 models (Base, Wide, and Deep) with 100 examples each. Each data example is associated with an $\ell_\infty$ norm $\epsilon$ and a target label for verification (referred to as a *property* to verify). The details of neural network structures and experimental setups can be found in Appendix C. We compare against multiple baselines for complete verification: (1) BaBSR [7], a basic BaB and LP based verifier; (2) MIPplanet [15], a customized MIP solver for NN verification where unstable ReLU neurons are randomly selected for splitting; (3) ERAN [35, 33, 36, 34], an abstract interpretation based verifier which performs well on this benchmark in VNN-COMP 2020; (4) GNN-Online [23], a BaB and LP based verifiers using a learned Graph Neural Network (GNN) to guide the ReLU splits; (5) BDD+ BaBSR [6], a verification framework based on Lagrangian decomposition on GPUs (BDD+) with BaBSR branching strategy; (6) OVAL (BDD+ GNN) [6, 23], a strong verifier in VNN-COMP 2020 using BDD+ with GNN guiding the ReLU splits; (7) A.set BaBSR and (8) Big-M+A.set BaBSR [10], very recent dual-space verifiers on GPUs with a tighter linear relaxation than triangle LP relaxations; (9) Fast-and-Complete [45],

Table 1: Average runtime and average number of branches on three CIFAR-10 models over 100 properties.

| Method | CIFAR-10 Base | | | CIFAR-10 Wide | | | CIFAR-10 Deep | | |
|---|---|---|---|---|---|---|---|---|---|
| | time(s) | branches | %timeout | time(s) | branches | %timeout | time(s) | branches | %timeout |
| BaBSR [7] | 2367.78 | 1020.55 | 36.00 | 2871.14 | 812.65 | 49.00 | 2750.75 | 401.28 | 39.00 |
| MIPplanet [15] | 2849.69 | - | 68.00 | 2417.53 | - | 46.00 | 2302.25 | - | 40.00 |
| ERAN* [35, 33, 36, 34] | 805.94 | - | 5.00 | 632.20 | - | 9.00 | 545.72 | - | 0.00 |
| GNN-online [23] | 1794.85 | 565.13 | 33.00 | 1367.38 | 372.74 | 15.00 | 1055.33 | 131.85 | 4.00 |
| BDD+ BaBSR [6] | 807.91 | 195480.14 | 20.00 | 505.65 | 74203.11 | 10.00 | 266.28 | 12722.74 | 4.00 |
| OVAL (BDD+ GNN)* [6, 23] | 662.17 | 67938.38 | 16.00 | 280.38 | 17895.94 | 6.00 | 94.69 | 1990.34 | 1.00 |
| A.set BaBSR [10] | 381.78 | 12004.60 | 7.00 | 165.51 | 2233.10 | 3.00 | 190.28 | 2491.55 | 2.00 |
| BigM+A.set BaBSR [10] | 390.44 | 11938.75 | 7.00 | 172.65 | 4050.59 | 3.00 | 177.22 | 3275.25 | 2.00 |
| Fast-and-Complete [45] | 695.01 | 119522.65 | 17.00 | 495.88 | 80519.85 | 9.00 | 105.64 | 2455.11 | 1.00 |
| BaDNB (BDD+ FSB)[11] | 309.29 | 38239.04 | 7.00 | 165.53 | 11214.44 | 4.00 | 10.50 | 368.16 | 0.00 |
| $\beta$-CROWN BaBSR | 226.06 | 509608.50 | 6.00 | 118.26 | 217691.24 | 3.00 | 6.12 | 204.66 | 0.00 |
| $\beta$-CROWN FSB | **118.23** | 208018.21 | **3.00** | **78.32** | 116912.57 | **2.00** | **5.69** | 41.12 | **0.00** |

* OVAL (BDD+ GNN) and ERAN results are from VNN-COMP 2020 report [22]. Other results were reported by their authors.

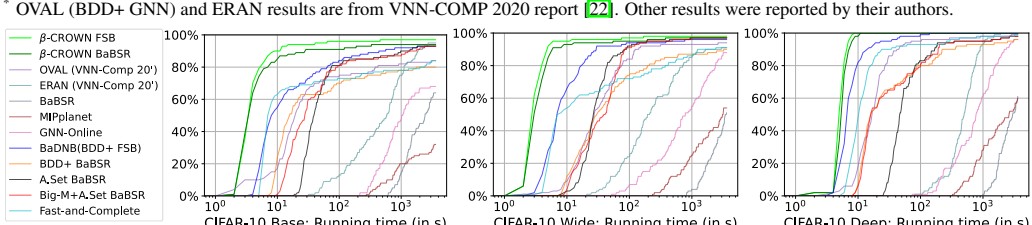

Figure 1: Percentage of solved properties with growing running time. $\beta$-CROWN FSB (light green) and $\beta$-CROWN BaBSR (dark green) clearly lead in all 3 settings and solve over 90% properties within 10 seconds.

which uses CROWN (LiRPA) on GPUs as the incomplete verifier in BaB without neuron split constraints; (10) BaDNB (BDD+ FSB) [11], a concurrent state-of-the-art complete verifier, using BDD+ on GPUs with FSB branching strategy. $\beta$-CROWN BaB can use either BaBSR or FSB branching heuristic, and we include both in evaluation. All methods use a 1 hour timeout threshold.

We report the average verification time and branch numbers in Table 1 and plot the percentage of solved properties over time in Figure 1. $\beta$-CROWN FSB achieves the fastest average running time compared to all other baselines with minimal timeouts, and also clearly leads on the cactus plot. When using a weaker branching heuristic, $\beta$-CROWN BaBSR still outperforms all literature baselines, including very recent ones such as A.set BaBSR [10], Fast-and-Complete [45] and BaDNB [11]. Our benefits are more clearly shown in Figure 1, where we solve over 90% examples under 10s and most other verifiers can verify much less or none of the properties within 10s. We see a 2 to 3 orders of magitudes speedup in Figure 1 compared to CPU based verifiers such as MIPplanet and BaBSR.

## 4.2 Comparison to Incomplete Verifiers

**Verified accuracy.** In Table 2, we compare against a few representative and strong incomplete verifiers on 5 convolutional networks and 4 MLP networks for MNIST and CIFAR-10 under the same set of 1000 images and perturbation $\epsilon$ as reported in [35, 37, 26]. Among the baselines, kPoly [35], OptC2V [37] and PRIMA [26] utilize state-of-the-art multi-neuron linear relaxation for ReLUs and can bypass the single-neuron convex relaxation barrier [30], and are among the strongest incomplete verifiers. $\beta$-CROWN FSB achieves better verified accuracy on all models using a similar or less amount of time. Some models, such as MNIST ConvBig and CIFAR ConvBig, are quite challenging - the verified accuracy obtained by $\beta$-CROWN FSB is close to the upper bound found via PGD attack.

To make more comprehensive evaluations, in Table 3 we further compare against a state-of-the-art semidefinite programming (SDP) based verifier, SDP-FO [9], on one MNIST and six CIFAR-10 models reported in their paper. The models were trained using adversarial training, which posed a challenge for verification [28]. The SDP formulation can be tighter than linear relaxation based ones, but it is computationally expensive - SDP-FO takes 2 to 3 hours to converge on one GPU for verifying a single property, resulting 5,400 GPU hours to verify 200 testing images with 10 labels each. Due to resource limitations, we directly quote SDP-FO results from [9] on the same set of models. We evaluate verified accuracy on the same set of 200 test images for other baselines. We include a concurrent work PRIMA [26], the strongest multi-neuron linear relaxation baseline in Table 2, which generally outperforms kPoly and OptC2V. Table 3 shows that overall we are 3 orders of magnitude faster than SDP-FO while still achieving consistently higher verified accuracy on average.

**Tightness of verification.** In Figure 2, we compare the tightness of verification bounds against SDP-FO on two adversarially trained networks from [9]. Specifically, we use the verification objective

Table 2: **Verified accuracy (%)** and avg. time (s) of 1000 images evaluated on the ERAN models in [35, 37, 26]. kPoly, OptC2V and PRIMA are strong incomplete verifiers that can break the convex relaxation barrier [30]. The average time reported by us excludes examples that are classified incorrectly.

| Dataset | Model (Same settings as [35, 37, 26]) | CROWN/DeepPoly* [36] Verified% | Time (s) | kPoly [35] Ver.% | Time(s) | OptC2V [37] Ver.% | Time(s) | PRIMA† [26] Ver.% | Time(s) | β-CROWN FSB Ver.% | Time(s) | Upper bound |
|---|---|---|---|---|---|---|---|---|---|---|---|---|
| MNIST | MLP $5 \times 100$‡ | 16.0 | 0.7 | 44.1 | 307 | 42.9 | 137 | 51.0 | 159 | **69.9** | 102 | 84.2 |
| | MLP $8 \times 100$ | 18.2 | 1.4 | 36.9 | 171 | 38.4 | 759 | 42.8 | 301 | **62.0** | 103 | 82.0 |
| | MLP $5 \times 200$ | 29.2 | 2.4 | 57.4 | 187 | 60.1 | 403 | 69.0 | 224 | **77.4** | 86 | 90.1 |
| | MLP $8 \times 200$ | 25.9 | 5.6 | 50.6 | 464 | 52.8 | 3451 | 62.4 | 395 | **73.5** | 95 | 91.1 |
| | ConvSmall | 15.8 | 3 | 34.7 | 477 | 43.6 | 55 | 59.8 | 42 | **72.7** | 7.0 | 73.2 |
| | ConvBig | 71.1 | 21 | 73.6 | 40 | 77.1 | 102 | 77.5 | 11 | **79.3** | 3.1 | 80.4 |
| CIFAR | ConvSmall | 35.9 | 4 | 39.9 | 86 | 39.8 | 105 | 44.6 | 13 | **46.3** | 6.8 | 48.1 |
| | ConvBig | 42.1 | 43 | 45.9 | 346 | No public code | | 48.3 | 176 | **51.6** | 15.3 | 55.0 |
| | ResNet | 24.1 | 1 | 24.5 | 91 | cannot run | | **24.8** | 1.7 | **24.8** | 1.6 | 24.8 |

* CROWN/DeepPoly evaluated on CPU. † PRIMA is a concurrent work and its results are from [26] (Oct 26, 2021 version), except that ResNet results are from personal communications with the authors due to a different input normalization used. ‡ Because these MLP models are fairly small, some of their intermediate layer bounds are computed by mixed integer programming (MIP) using 80% time budget before branch and bound starts and β-CROWN FSB is used during the branch and bound process. We find that tighter intermediate bounds by MIP is beneficial for these small MLP models.

Table 3: **Verified accuracy (%)** and avg. per-example verification time (s) on 7 models from SDP-FO [9]. CROWN/DeepPoly are fast but loose bound propagation based methods, and they cannot be improved with more running time. SDP-FO uses stronger semidefinite relaxations, which can be very slow and sometimes has convergence issues. PRIMA, a concurrent work, is the state-of-the-art relaxation barrier breaking method; we did not include kPoly and OptC2V because they are weaker than PRIMA (see Table 2).

| Dataset | Model $\epsilon = 0.3$ and $\epsilon = 2/255$ | CROWN/DeepPoly Verified% | Time (s) | SDP-FO [9]* Ver.% | Time(s) | PRIMA [26] Ver.% | Time(s) | β-CROWN FSB Ver.% | Time(s) | Upper bound |
|---|---|---|---|---|---|---|---|---|---|---|
| MNIST | CNN-A-Adv | 1.0 | 0.1 | 43.4 | >20h | 44.5 | 135.9 | **70.5** | 21.1 | 76.5 |
| CIFAR | CNN-B-Adv | 21.5 | 0.5 | 32.8 | >25h | 38.0 | 343.6 | **46.5** | 32.2 | 65.0 |
| | CNN-B-Adv-4 | 43.5 | 0.9 | 46.0 | >25h | 53.5 | 43.8 | **54.0** | 11.6 | 63.5 |
| | CNN-A-Adv | 35.5 | 0.6 | 39.6 | >25h | 41.5 | 4.8 | **44.0** | 5.8 | 50.0 |
| | CNN-A-Adv-4 | 41.5 | 0.7 | 40.0 | >25h | 45.0 | 4.9 | **46.0** | 5.6 | 49.5 |
| | CNN-A-Mix | 23.5 | 0.4 | 39.6 | >25h | 37.5 | 34.3 | **41.5** | 49.6 | 53.0 |
| | CNN-A-Mix-4 | 38.0 | 0.5 | 47.8 | >25h | 48.5 | 7.0 | **50.5** | 5.9 | 57.5 |

* SDP-FO results are directly from their paper due to its very long running time (>20h per example). † PRIMA experiments were done using commit 396dc7a, released on June 4, 2021. PRIMA and β-CROWN FSB results are on the same set of 200 examples (first 200 examples of CIFAR-10 dataset) and we don't run verifiers on examples that are classified incorrectly or can be attacked by a 200-step PGD. β-CROWN uses 1 GPU and 1 CPU; PRIMA uses 1 GPU and 20 CPUs.

$f(x) := z_y^{(L)}(x) - z_{y'}^{(L)}(x)$, where $z^{(L)}$ is the logit layer output, $y$ and $y'$ are the true label and the runner-up label. For each test image, a 200-step PGD attack [24] provides an adversarial upper bound $\overline{f}$ of the optimal objective: $f^* \leq \overline{f}$. Verifiers, on the other hand, can provide a verified lower bound $\underline{f} \leq f^*$. Bounds from tighter verification methods lie closer to line $y = x$ in Figure 2. Figure 2 shows that on both PGD adversarially trained networks, β-CROWN FSB consistently outperforms SDP-FO for all 100 random test images. Importantly, for each point on the plots, β-CROWN FSB needs 3 minutes while SDP-FO needs 178 minutes on average. LP verifier with triangle relaxations produces much looser bounds than β-CROWN FSB and SDP-FO. Additional results are in Appendix C.2.

**VNN-COMP 2021 results.** We encourage the readers to checkout the report of the Second International Verification of Neural Networks Competition (VNN-COMP 2021) [3] with 9 additional benchmarks and 12 competing methods evaluated in a standardized testing environment on AWS. Our entry $\alpha,\beta$-CROWN is based on the β-CROWN algorithm in this work and uses the same codebase.

## 5 Related Work

Many early complete verifiers for neural networks relied on existing solvers such as MILP or SMT solvers [20, 15, 19, 12, 38] and were limited to very small problem instances. Branch and bound (BaB)

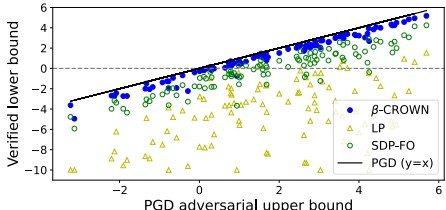
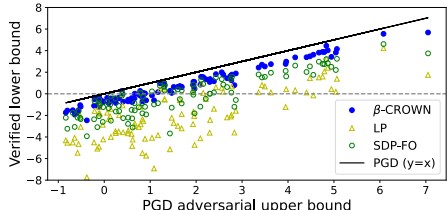

(a) MNIST CNN-A-Adv, runner-up targets, $\epsilon = 0.3$      (b) CIFAR CNN-B-Adv, runner-up targets, $\epsilon = 2/255$

Figure 2: Verified lower bound v.s. PGD adversarial upper bound. A lower bound closer to the upper bound (closer to the line $y = x$) is better. β-CROWN FSB uses 3mins while SDP-FO needs 2 to 3 hours per point.

based method was proposed to better exploit the network structure using LP-based incomplete verifier for bounding and ReLU splits for branching [8, 40, 23, 5]. Besides branching on ReLU neurons, input domain branching was also considered in [41, 29, 1] but limited by input dimensions [8].

Recently, a few approaches have been proposed to use efficient iterative solvers or bound propagation methods on GPUs without relying on LP solvers. Bunel et al. [6] decomposed the verification problem layer by layer, solved each layer in a closed form on GPUs, and used Lagrangian to enforce consistency between layers. However, their formulation only has the same power as LP and needs many iterations to converge. De Palma et al. [10] used a dual-space verifier with a linear relaxation [2, 37] tighter than triangle LP relaxation, but in most settings the extra computational costs and difficulties for an efficient implementation offset its benefits (more discussions in section B.2). A concurrent work BaDNB [11] proposed a new branching strategy, filtered smart branching (FSB), combined with Lagrangian decomposition to get better verification performance. Xu et al. [44] used CROWN as a massively paralleled incomplete solver on GPUs for complete verification, but it cannot handle neuron split constraints, leading to suboptimal efficiency and high timeout rates.

For incomplete verification, Salman et al. [30] shows the inherent limitation of using per-neuron convex relaxations for verification problems. Singh et al. [35] and Müller et al. [26] broke this barrier by considering constraints involving multiple ReLU neurons; Tjandraatmadja et al. [37] proposed to relax a linear layer with a ReLU neuron together using a strong mixed-integer programming formulation [1]. SDP based relaxations [28, 16, 14] typically produce tight bounds but with significantly higher cost. The most recent GPU based SDP verifier [9] is still relatively slow and can take 2 hours to verify a single image. In this work, we impose neuron split constraints using $\beta$-CROWN and combine it with branch and bound done in parallel on GPUs. Although for each subdomain in BaB, $\beta$-CROWN is still subject to the convex relaxation barrier, the efficiency of $\beta$-CROWN BaB allows it to quickly explore a very large number of subdomains and outperform existing convex barrier breaking incomplete verifiers under many scenarios in both runtime and tightness.

Additionally, another line of works train networks to enhance verified accuracy, typically using cheap incomplete verifiers at training time [42, 39, 25, 43, 18, 25, 47, 4, 32]. Traditionally only these verification-customized networks can have reasonable verified accuracy, while $\beta$-CROWN BaB can also give non-trivial verified accuracy on relatively large networks agnostic to verification.

## 6   Conclusion

We proposed $\beta$-CROWN, a new bound propagation method that can fully encode the neuron split constraints introduced in BaB, which clearly leads in both complete and incomplete verification settings. The success of $\beta$-CROWN comes from a few factors: (1) In Section 3.1, we show that $\beta$-CROWN is an GPU-friendly bound propagation algorithm *significantly faster than LP solvers*. (2) In Section 3.2, we show that $\beta$-CROWN is solving an equivalent problem of the LP verifier *with neuron split constraints*. (3) In Section 3.3, we show that $\beta$-CROWN can jointly optimize intermediate layer bounds and *achieve tighter bounds than typical LP verifiers* using fixed intermediate layer bounds.

**Limitations.**   Our verifier has several limitations which are commonly shared by most existing BaB-based complete verifiers. First, we focused on ReLU which can be split into two linear cases. For other non-piecewise linear activation functions, although it is still possible to conduct branch and bound, it is difficult to guarantee completeness. Second, we discussed only the norm perturbations for input domains. In practice, the threat model may involve complicated and nonconvex perturbation specifications. Third, although our GPU accelerated verifier outperforms existing ones, all BaB based verifiers, including ours, are still limited to relatively small models far from the ImageNet scale. Finally, we have only demonstrated robustness verification of image classification tasks, and generalizing it to give verification guarantees for other tasks such as robust deep reinforcement learning [27, 48, 49] is an interesting direction for future work.

**Societal Impact**   NNs have been used in an increasingly wide range of real-world applications and play an important role in artificial intelligence (AI). The trustworthiness and robustness of NNs have become crucial factors since AI plays an important role in modern society. $\beta$-CROWN is a strong neural network verifier which can be used to check certain properties of neural networks, which can be helpful for guaranteeing the robustness, correctness, and fairness of NNs in applications that can directly or indirectly impact human life. We believe our work has overall positive societal impacts, although it may potentially be misused to identify the weakness of NNs and guide attacks.

## Acknowledgement

This work is supported by NSF grant CNS18-01426; an ARL Young Investigator (YIP) award; an NSF CAREER award; a Google Faculty Fellowship; a Capital One Research Grant; and a J.P. Morgan Faculty Award; Air Force Research Laboratory under FA8750-18-2-0058; NSF IIS-1901527, NSF IIS-2008173 and NSF CAREER-2048280; and NSF CNS-1932351. Huan Zhang is supported by funding from the Bosch Center for Artificial Intelligence.

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
