# A Proofs for $\beta$-CROWN

## A.1 Proofs for deriving $\beta$-CROWN using bound propagation

Lemma 2.1 is from part of the proof of the main theorem in Zhang et al. [46]. Here we present it separately to use it as an useful subprocedure for our later proofs.

**Lemma 2.1** (Relaxation of a ReLU layer in CROWN). *Given two vectors* $w, v \in \mathbb{R}^d, \mathbf{l} \leq v \leq \mathbf{u}$ *(element-wise), we have*

$$w^\top \mathrm{ReLU}(v) \geq w^\top \mathbf{D} v + b',$$

*where* $\mathbf{D}$ *is a diagonal matrix defined as:*

$$\mathbf{D}_{j,j} = \begin{cases} 1, & \text{if } \mathbf{l}_j \geq 0 \\ 0, & \text{if } \mathbf{u}_j \leq 0 \\ \boldsymbol{\alpha}_j, & \text{if } \mathbf{u}_j > 0 > \mathbf{l}_j \text{ and } w_j \geq 0 \\ \frac{\mathbf{u}_j}{\mathbf{u}_j - \mathbf{l}_j}, & \text{if } \mathbf{u}_j > 0 > \mathbf{l}_j \text{ and } w_j < 0, \end{cases} \tag{1}$$

$0 \leq \boldsymbol{\alpha}_j \leq 1$ *are free variables,* $b' = w^\top \underline{\mathbf{b}}$ *and each element in* $\underline{\mathbf{b}}$ *is*

$$\underline{\mathbf{b}}_j = \begin{cases} 0, & \text{if } \mathbf{l}_j > 0 \text{ or } \mathbf{u}_j \leq 0 \\ 0, & \text{if } \mathbf{u}_j > 0 > \mathbf{l}_j \text{ and } w_j \geq 0 \\ -\frac{\mathbf{u}_j \mathbf{l}_j}{\mathbf{u}_j - \mathbf{l}_j}, & \text{if } \mathbf{u}_j > 0 > \mathbf{l}_j \text{ and } w_j < 0. \end{cases} \tag{2}$$

*Proof.* For the $j$-th ReLU neuron, if $\mathbf{l}_j \geq 0$, then $\mathrm{ReLU}(v_j) = v_j$; if $\mathbf{u}_j < 0$, then $\mathrm{ReLU}(v_j) = 0$. For the case of $\mathbf{l}_j < 0 < \mathbf{u}_j$, the ReLU function can be linearly upper and lower bounded within this range:

$$\boldsymbol{\alpha}_j v_j \leq \mathrm{ReLU}(v_j) \leq \frac{\mathbf{u}_j}{\mathbf{u}_j - \mathbf{l}_j} (v_j - \mathbf{l}_j) \quad \forall \mathbf{l}_j \leq v_j \leq \mathbf{u}_j$$

where $0 \leq \boldsymbol{\alpha}_j \leq 1$ is a free variable - any value between 0 and 1 produces a valid lower bound. To lower bound $w^\top \mathrm{ReLU}(v) = \sum_j w_j \mathrm{ReLU}(v_j)$, for each term in this summation, we take the lower bound of $\mathrm{ReLU}(v_j)$ if $w_j$ is positive and take the upper bound of $\mathrm{ReLU}(v_j)$ if $w_j$ is negative (reflected in the definitions of $\mathbf{D}$ and $\underline{\mathbf{b}}$). This conservative choice allows us to always obtain a lower bound $\forall \mathbf{l} \leq v \leq \mathbf{u}$:

$$\sum_j w_j \mathrm{ReLU}(v_j) \geq \sum_j w_j \left( \mathbf{D}_{j,j} v_j + \underline{\mathbf{b}}_j \right) = w^\top \mathbf{D} v + w^\top \underline{\mathbf{b}} = w^\top \mathbf{D} v + b'$$

where $\mathbf{D}_{j,j}$ and $\underline{\mathbf{b}}_j$ are defined in Eq. 1 and Eq. 2 representing the lower or upper bounds of ReLU. $\qquad\square$

Before proving our main theorem (Theorem 3.1), we first define matrix $\boldsymbol{\Omega}$, which is the product of a series of model weights $\mathbf{W}$ and "weights" for relaxed ReLU layers $\mathbf{D}$:

**Definition A.1.** *Given a set of matrices* $\mathbf{W}^{(2)}, \cdots, \mathbf{W}^{(L)}$ *and* $\mathbf{D}^{(1)}, \cdots, \mathbf{D}^{(L-1)}$, *we define a recursive function* $\boldsymbol{\Omega}(k, i)$ *for* $1 \leq i \leq k \leq L$ *as*

$$\boldsymbol{\Omega}(i, i) = \boldsymbol{I}, \ \boldsymbol{\Omega}(k+1, i) = \mathbf{W}^{(k+1)} \mathbf{D}^{(k)} \boldsymbol{\Omega}(k, i)$$

For example, $\boldsymbol{\Omega}(3, 1) = \mathbf{W}^{(3)} \mathbf{D}^{(2)} \mathbf{W}^{(2)} \mathbf{D}^{(1)}$, $\boldsymbol{\Omega}(5, 2) = \mathbf{W}^{(5)} \mathbf{D}^{(4)} \mathbf{W}^{(4)} \mathbf{D}^{(3)} \mathbf{W}^{(3)} \mathbf{D}^{(2)}$. Now we present our main theorem with each term explicitly written:

**Theorem 3.1** ($\beta$-CROWN bound). *Given a $L$-layer neural network* $f(x) : \mathbb{R}^{d_0} \to \mathbb{R}$ *with weights* $\mathbf{W}^{(i)}$, *biases* $\mathbf{b}^{(i)}$, *pre-ReLU bounds* $\mathbf{l}^{(i)} \leq z^{(i)} \leq \mathbf{u}^{(i)}$ ($1 \leq i \leq L$), *input constraint* $\mathcal{C}$ *and split constraint* $\mathcal{Z}$. *We have*

$$\min_{x \in \mathcal{C}, z \in \mathcal{Z}} f(x) \geq \max_{\boldsymbol{\beta} \geq 0} \min_{x \in \mathcal{C}} (\boldsymbol{a} + \mathbf{P}\boldsymbol{\beta})^\top x + \mathbf{q}^\top \boldsymbol{\beta} + c, \tag{3}$$

*where* $\mathbf{P} \in \mathbb{R}^{d_0 \times (\sum_{i=1}^{L-1} d_i)}$ *is a matrix containing blocks* $\mathbf{P} := \begin{bmatrix} \mathbf{P}_1^\top & \mathbf{P}_2^\top & \cdots & \mathbf{P}_{L-1}^\top \end{bmatrix}$, $\mathbf{q} \in \mathbb{R}^{\sum_{i=1}^{L-1} d_i}$ *is a vector* $\mathbf{q} := \begin{bmatrix} \mathbf{q}_1^\top & \cdots & \mathbf{q}_{L-1}^\top \end{bmatrix}^\top$, *and each term is defined as:*

$$\boldsymbol{a} = \left[ \boldsymbol{\Omega}(L, 1) \mathbf{W}^{(1)} \right]^\top \in \mathbb{R}^{d_0 \times 1} \tag{4}$$

$$\mathbf{P}_i = \mathbf{S}^{(i)}\mathbf{\Omega}(i,1)\mathbf{W}^{(1)} \in \mathbb{R}^{d_i \times d_0}, \quad \forall\, 1 \le i \le L-1 \tag{5}$$

$$\mathbf{q}_i = \sum_{k=1}^{i} \mathbf{S}^{(i)}\mathbf{\Omega}(i,k)\mathbf{b}^{(k)} + \sum_{k=2}^{i} \mathbf{S}^{(i)}\mathbf{\Omega}(i,k)\mathbf{W}^{(k)}\underline{\mathbf{b}}^{(k-1)} \in \mathbb{R}^{d_i}, \quad \forall\, 1 \le i \le L-1 \tag{6}$$

$$c = \sum_{i=1}^{L} \mathbf{\Omega}(L,i)\mathbf{b}^{(i)} + \sum_{i=2}^{L} \mathbf{\Omega}(L,i)\mathbf{W}^{(i)}\underline{\mathbf{b}}^{(i-1)} \tag{7}$$

*diagonal matrices $\mathbf{D}^{(i)}$ and vector $\underline{\mathbf{b}}^{(i)}$ are determined by the relaxation of ReLU neurons, and $\mathbf{A}^{(i)} \in \mathbb{R}^{1 \times d_i}$ represents the linear relationship between $f(x)$ and $\hat{z}^{(i)}$. $\mathbf{D}^{(i)}$ and $\underline{\mathbf{b}}^{(i)}$ depend on $\mathbf{A}^{(i)}$, $\mathbf{l}^{(i)}$ and $\mathbf{u}^{(i)}$:*

$$\mathbf{D}_{j,j}^{(i)} = \begin{cases} 1, & if\, \mathbf{l}_j^{(i)} \ge 0 \ or \ j \in \mathcal{Z}^{+(i)} \\ 0, & if\, \mathbf{u}_j^{(i)} \le 0 \ or \ j \in \mathcal{Z}^{-(i)} \\ \boldsymbol{\alpha}_j, & if\, \mathbf{u}_j^{(i)} > 0 > \mathbf{l}_j^{(i)} \ and \ j \notin \mathcal{Z}^{+(i)} \cup \mathcal{Z}^{-(i)} \ and \ \mathbf{A}_{1,j}^{(i)} \ge 0 \\ \frac{\mathbf{u}_j}{\mathbf{u}_j - \mathbf{l}_j}, & if\, \mathbf{u}_j^{(i)} > 0 > \mathbf{l}_j^{(i)} \ and \ j \notin \mathcal{Z}^{+(i)} \cup \mathcal{Z}^{-(i)} \ and \ \mathbf{A}_{1,j}^{(i)} < 0 \end{cases} \tag{8}$$

$$\underline{\mathbf{b}}_j^{(i)} = \begin{cases} 0, & if\, \mathbf{l}_j^{(i)} > 0 \ or \ \mathbf{u}_j^{(i)} \le 0 \ or \ j \in \mathcal{Z}^{+(i)} \cup \mathcal{Z}^{-(i)} \\ 0, & if\, \mathbf{u}_j^{(i)} > 0 > \mathbf{l}_j^{(i)} \ and \ j \notin \mathcal{Z}^{+(i)} \cup \mathcal{Z}^{-(i)} \ and \ \mathbf{A}_{1,j}^{(i)} \ge 0 \\ -\frac{\mathbf{u}_j^{(i)}\mathbf{l}_j^{(i)}}{\mathbf{u}_j^{(i)} - \mathbf{l}_j^{(i)}}, & if\, \mathbf{u}_j^{(i)} > 0 > \mathbf{l}_j^{(i)} \ and \ j \notin \mathcal{Z}^{+(i)} \cup \mathcal{Z}^{-(i)} \ and \ \mathbf{A}_{1,j}^{(i)} < 0 \end{cases} \tag{9}$$

$$\mathbf{A}^{(i)} = \begin{cases} \mathbf{W}^{(L)}, & i = L-1 \\ (\mathbf{A}^{(i+1)}\mathbf{D}^{(i+1)} + \boldsymbol{\beta}^{(i+1)\top}\mathbf{S}^{(i+1)})\mathbf{W}^{(i+1)}, & 0 \le i \le L-2 \end{cases} \tag{10}$$

*Proof.* We prove this theorem by induction: assuming we know the bounds with respect to layer $\hat{z}^{(m)}$, we derive bounds for $\hat{z}^{(m-1)}$ until we reach $m = 0$ and by definition $\hat{z}^{(0)} = x$. We first define a set of matrices and vectors $\boldsymbol{a}^{(m)}, \mathbf{P}^{(m)}, \mathbf{q}^{(m)}, c^{(m)}$, where $\mathbf{P}^{(m)} \in \mathbb{R}^{d_m \times (\sum_{i=m+1}^{L-1} d_i)}$ is a matrix containing blocks $\mathbf{P} := \left[ \mathbf{P}_{m+1}^{(m)\top} \cdots \mathbf{P}_{L-1}^{(m)\top} \right]$, $\mathbf{q} \in \mathbb{R}^{\sum_{i=m+1}^{L-1} d_i}$ is a vector $\mathbf{q} := \left[ \mathbf{q}_{m+1}^{(m)\top} \cdots \mathbf{q}_{L-1}^{(m)\top} \right]^\top$, and each term is defined as:

$$\boldsymbol{a}^{(m)} = \left[ \mathbf{\Omega}(L, m+1)\mathbf{W}^{(m+1)} \right]^\top \in \mathbb{R}^{d_m \times 1} \tag{11}$$

$$\mathbf{P}_i^{(m)} = \mathbf{S}^{(i)}\mathbf{\Omega}(i, m+1)\mathbf{W}^{(m+1)} \in \mathbb{R}^{d_i \times d_m}, \quad \forall\, m+1 \le i \le L-1 \tag{12}$$

$$\mathbf{q}_i^{(m)} = \sum_{k=m+1}^{i} \mathbf{S}^{(i)}\mathbf{\Omega}(i,k)\mathbf{b}^{(k)} + \sum_{k=m+2}^{i} \mathbf{S}^{(i)}\mathbf{\Omega}(i,k)\mathbf{W}^{(k)}\underline{\mathbf{b}}^{(k-1)} \in \mathbb{R}^{d_m}, \quad \forall\, m+1 \le i \le L-1 \tag{13}$$

$$c^{(m)} = \sum_{i=m+1}^{L} \mathbf{\Omega}(L,i)\mathbf{b}^{(i)} + \sum_{i=m+2}^{L} \mathbf{\Omega}(L,i)\mathbf{W}^{(i)}\underline{\mathbf{b}}^{(i-1)} \tag{14}$$

and we claim that

$$\min_{\substack{x \in \mathcal{C} \\ z \in \mathcal{Z}}} f(x) \ge \max_{\tilde{\boldsymbol{\beta}}^{(m+1)} \ge 0} \min_{\substack{x \in \mathcal{C} \\ \hat{z}^{(m)} \in \tilde{\mathcal{Z}}^{(m)}}} (\boldsymbol{a}^{(m)} + \mathbf{P}^{(m)}\tilde{\boldsymbol{\beta}}^{(m+1)})^\top \hat{z}^{(m)} + \mathbf{q}^{(m)\top}\tilde{\boldsymbol{\beta}}^{(m+1)} + c^{(m)} \tag{15}$$

where $\tilde{\boldsymbol{\beta}}^{(m+1)} := \left[ \boldsymbol{\beta}^{(m+1)\top} \cdots \boldsymbol{\beta}^{(L-1)\top} \right]^\top$ concatenating all $\boldsymbol{\beta}^{(i)}$ variables up to layer $m+1$.

For the base case $m = L - 1$, we simply have

$$\min_{x \in \mathcal{C}, z \in \mathcal{Z}} f(x) = \min_{x \in \mathcal{C}, z \in \mathcal{Z}} \mathbf{W}^{(L)} \hat{z}^{(L-1)} + \mathbf{b}^{(L)}.$$

No maximization is needed and $\boldsymbol{a}^{(m)} = \left[ \boldsymbol{\Omega}(L, L) \mathbf{W}^{(L)} \right]^{\top} = \mathbf{W}^{(L)\top}$, $c^{(m)} = \sum_{i=L}^{L} \boldsymbol{\Omega}(L, i) \mathbf{b}^{(i)} = \mathbf{b}^{(L)}$. Other terms are zero.

In Section 3.1 we have shown the intuition of the proof by demonstrating how to derive the bounds from layer $\hat{z}^{(L-1)}$ to $\hat{z}^{(L-2)}$. The case for $m = L - 2$ is presented in Eq. 6.

Now we show the induction from $\hat{z}^{(m)}$ to $\hat{z}^{(m-1)}$. Starting from Eq. 15, since $\hat{z}^{(m)} = \text{ReLU}(z^{(m)})$ we apply Lemma 2.1 by setting $w = \left[ \boldsymbol{a}^{(m)} + \mathbf{P}^{(m)} \tilde{\boldsymbol{\beta}}^{(m+1)} \right]^{\top} := \mathbf{A}^{(m)}$. It is easy to show that $\mathbf{A}^{(m)}$ can also be equivalently and recursively defined in Eq. 10 (see Lemma A.2). Based on Lemma 2.1 we have $\mathbf{D}^{(m)}$ and $\underline{\mathbf{b}}^{(m)}$ defined as in Eq. 8 and Eq. 9, so Eq. 15 becomes

$$\min_{\substack{x \in \mathcal{C} \\ z \in \mathcal{Z}}} f(x) \geq \max_{\tilde{\boldsymbol{\beta}}^{(m+1)} \geq 0} \min_{\substack{x \in \mathcal{C} \\ z^{(m)} \in \tilde{\mathcal{Z}}^{(m)}}} (\boldsymbol{a}^{(m)} + \mathbf{P}^{(m)} \tilde{\boldsymbol{\beta}}^{(m+1)})^{\top} \mathbf{D}^{(m)} z^{(m)}$$
$$+ (\boldsymbol{a}^{(m)} + \mathbf{P}^{(m)} \tilde{\boldsymbol{\beta}}^{(m+1)})^{\top} \underline{\mathbf{b}}^{(m)} + \mathbf{q}^{(m)\top} \tilde{\boldsymbol{\beta}}^{(m+1)} + c^{(m)} \tag{16}$$

Note that when we apply Lemma 2.1, for $j \in \mathcal{Z}^{+(i)}$ (positive split) we simply treat the neuron $j$ as if $\mathbf{l}_j^{(i)} \geq 0$, and for $j \in \mathcal{Z}^{-(i)}$ (negative split) we simply treat the neuron $j$ as if $\mathbf{u}_j^{(i)} \leq 0$. Now we add the multiplier $\boldsymbol{\beta}^{(m)}$ to $z^{(m)}$ to enforce per-neuron split constraints:

$$\min_{\substack{x \in \mathcal{C} \\ z \in \mathcal{Z}}} f(x) \geq \max_{\tilde{\boldsymbol{\beta}}^{(m+1)} \geq 0} \min_{\substack{x \in \mathcal{C} \\ z^{(m-1)} \in \tilde{\mathcal{Z}}^{(m-1)}}} \max_{\boldsymbol{\beta}^{(m)} \geq 0} (\boldsymbol{a}^{(m)} + \mathbf{P}^{(m)} \tilde{\boldsymbol{\beta}}^{(m+1)})^{\top} \mathbf{D}^{(m)} z^{(m)} + \boldsymbol{\beta}^{(m)\top} \mathbf{S}^{(m)} z^{(m)}$$
$$+ (\boldsymbol{a}^{(m)} + \mathbf{P}^{(m)} \tilde{\boldsymbol{\beta}}^{(m+1)})^{\top} \underline{\mathbf{b}}^{(m)} + \mathbf{q}^{(m)\top} \tilde{\boldsymbol{\beta}}^{(m+1)} + c^{(m)}$$
$$\geq \max_{\tilde{\boldsymbol{\beta}}^{(m)} \geq 0} \min_{\substack{x \in \mathcal{C} \\ z^{(m-1)} \in \tilde{\mathcal{Z}}^{(m-1)}}} (\boldsymbol{a}^{(m)\top} \mathbf{D}^{(m)} + \tilde{\boldsymbol{\beta}}^{(m+1)\top} \mathbf{P}^{(m)\top} \mathbf{D}^{(m)} + \boldsymbol{\beta}^{(m)\top} \mathbf{S}^{(m)}) z^{(m)}$$
$$+ (\boldsymbol{a}^{(m)} + \mathbf{P}^{(m)} \tilde{\boldsymbol{\beta}}^{(m+1)})^{\top} \underline{\mathbf{b}}^{(m)} + \mathbf{q}^{(m)\top} \tilde{\boldsymbol{\beta}}^{(m+1)} + c^{(m)}$$

Similar to what we did in Eq. 5, we swap the min and max in the second inequality due to weak duality, such that every maximization on $\boldsymbol{\beta}^{(i)}$ is before min. Then, we substitute $\hat{z}^{(m)} = \mathbf{W}^{(m)} \hat{z}^{(m-1)} + \mathbf{b}^{(m)}$ and obtain:

$$\min_{\substack{x \in \mathcal{C} \\ z \in \mathcal{Z}}} f(x) \geq \max_{\tilde{\boldsymbol{\beta}}^{(m)} \geq 0} \min_{\substack{x \in \mathcal{C} \\ z^{(m-1)} \in \tilde{\mathcal{Z}}^{(m-1)}}} (\boldsymbol{a}^{(m)\top} \mathbf{D}^{(m)} + \tilde{\boldsymbol{\beta}}^{(m+1)\top} \mathbf{P}^{(m)\top} \mathbf{D}^{(m)} + \boldsymbol{\beta}^{(m)\top} \mathbf{S}^{(m)})^{\top} \mathbf{W}^{(m)} \hat{z}^{(m-1)}$$
$$+ (\boldsymbol{a}^{(m)\top} \mathbf{D}^{(m)} + \tilde{\boldsymbol{\beta}}^{(m+1)\top} \mathbf{P}^{(m)\top} \mathbf{D}^{(m)} + \boldsymbol{\beta}^{(m)\top} \mathbf{S}^{(m)})^{\top} \mathbf{b}^{(m)}$$
$$+ (\boldsymbol{a}^{(m)} + \mathbf{P}^{(m)} \tilde{\boldsymbol{\beta}}^{(m+1)})^{\top} \underline{\mathbf{b}}^{(m)} + \mathbf{q}^{(m)\top} \tilde{\boldsymbol{\beta}}^{(m+1)} + c^{(m)}$$
$$= \left[ \underbrace{\left[ \boldsymbol{a}^{(m)\top} \mathbf{D}^{(m)} \mathbf{W}^{(m)} \right]^{\top}}_{\boldsymbol{a}'} + \underbrace{(\tilde{\boldsymbol{\beta}}^{(m+1)\top} \mathbf{P}^{(m)\top} \mathbf{D}^{(m)} \mathbf{W}^{(m)} + \boldsymbol{\beta}^{(m)\top} \mathbf{S}^{(m)} \mathbf{W}^{(m)})}_{\mathbf{P}' \tilde{\boldsymbol{\beta}}^{(m)}} \right]^{\top} \hat{z}^{(m-1)}$$
$$+ \underbrace{\left( (\mathbf{P}^{(m)\top} \mathbf{D}^{(m)} \mathbf{b}^{(m)} + \mathbf{P}^{(m)\top} \underline{\mathbf{b}}^{(m)} + \mathbf{q}^{(m)})^{\top} \tilde{\boldsymbol{\beta}}^{(m+1)} + (\mathbf{S}^{(m)} \mathbf{b}^{(m)})^{\top} \boldsymbol{\beta}^{(m)} \right)}_{\mathbf{q}'^{\top} \tilde{\boldsymbol{\beta}}^{(m)}}$$
$$+ \underbrace{\boldsymbol{a}^{(m)\top} \mathbf{D}^{(m)} \mathbf{b}^{(m)} + \boldsymbol{a}^{(m)\top} \underline{\mathbf{b}}^{(m)} + c^{(m)}}_{c'}$$

Now we evaluate each term $\boldsymbol{a}'$, $\mathbf{P}'$, $\mathbf{q}'$ and $c'$ and show the induction holds. For $\boldsymbol{a}'$ and $\mathbf{q}'$ we have:

$$\boldsymbol{a}' = \left[\boldsymbol{a}^{(m)\top}\mathbf{D}^{(m)}\mathbf{W}^{(m)}\right]^{\top} = \left[\boldsymbol{\Omega}(L, m+1)\mathbf{W}^{(m+1)}\mathbf{D}^{(m)}\mathbf{W}^{(m)}\right]^{\top} = \left[\boldsymbol{\Omega}(L, m)\mathbf{W}^{(m)}\right]^{\top} = \boldsymbol{a}^{(m-1)}$$

$$c' = c^{(m)} + \boldsymbol{\Omega}(L, m+1)\mathbf{W}^{(m+1)}\mathbf{D}^{(m)}\mathbf{b}^{(m)} + \boldsymbol{\Omega}(L, m+1)\mathbf{W}^{(m+1)}\underline{\mathbf{b}}^{(m)}$$

$$= \sum_{i=m+1}^{L} \boldsymbol{\Omega}(L, i)\mathbf{b}^{(i)} + \sum_{i=m+2}^{L} \boldsymbol{\Omega}(L, i)\mathbf{W}^{(i)}\underline{\mathbf{b}}^{(i-1)} + \boldsymbol{\Omega}(L, m)\mathbf{b}^{(m)} + \boldsymbol{\Omega}(L, m+1)\mathbf{W}^{(m+1)}\underline{\mathbf{b}}^{(m)}$$

$$= \sum_{i=m}^{L} \boldsymbol{\Omega}(L, i)\mathbf{b}^{(i)} + \sum_{i=m+1}^{L} \boldsymbol{\Omega}(L, i)\mathbf{W}^{(i)}\underline{\mathbf{b}}^{(i-1)}$$

$$= c^{(m-1)}$$

For $\mathbf{P}' := \left[\mathbf{P}'_m{}^{\top} \quad \cdots \quad \mathbf{P}'_{L-1}{}^{\top}\right]$, we have a new block $\mathbf{P}'_m$ where

$$\mathbf{P}'_m = \mathbf{S}^{(m)}\mathbf{W}^{(m)} = \mathbf{S}^{(m)}\boldsymbol{\Omega}(m, m)\mathbf{W}^{(m)} = \mathbf{P}_m^{(m-1)}$$

for other blocks where $m + 1 \le i \le L - 1$,

$$\mathbf{P}'_i = \mathbf{P}_i^{(m)}\mathbf{D}^{(m)}\mathbf{W}^{(m)} = \mathbf{S}^{(i)}\boldsymbol{\Omega}(i, m+1)\mathbf{W}^{(m+1)}\mathbf{D}^{(m)}\mathbf{W}^{(m)} = \mathbf{S}^{(i)}\boldsymbol{\Omega}(i, m)\mathbf{W}^{(m)} = \mathbf{P}_i^{(m-1)}$$

For $\mathbf{q}' := \left[\mathbf{q}'_m{}^{\top} \quad \cdots \quad \mathbf{q}'_{L-1}{}^{\top}\right]$, we have a new block $\mathbf{q}'_m$ where

$$\mathbf{q}'_m = \mathbf{S}^{(m)}\mathbf{b}^{(m)} = \sum_{k=m}^{m} \mathbf{S}^{(i)}\boldsymbol{\Omega}(i, k)\mathbf{b}^{(i)} = \mathbf{q}_m^{(m-1)}$$

for other blocks where $m + 1 \le i \le L - 1$,

$$\mathbf{q}'_i = \mathbf{q}_i^{(m)} + \mathbf{P}^{(m)\top}\mathbf{D}^{(m)}\mathbf{b}^{(m)} + \mathbf{P}^{(m)\top}\underline{\mathbf{b}}^{(m)}$$

$$= \sum_{k=m+1}^{i} \mathbf{S}^{(i)}\boldsymbol{\Omega}(i, k)\mathbf{b}^{(k)} + \sum_{k=m+2}^{i} \mathbf{S}^{(i)}\boldsymbol{\Omega}(i, k)\mathbf{W}^{(k)}\underline{\mathbf{b}}^{(k-1)} + \mathbf{P}^{(m)\top}\mathbf{D}^{(m)}\mathbf{b}^{(m)} + \mathbf{P}^{(m)\top}\underline{\mathbf{b}}^{(m)}$$

$$= \sum_{k=m+1}^{i} \mathbf{S}^{(i)}\boldsymbol{\Omega}(i, k)\mathbf{b}^{(k)} + \sum_{k=m+2}^{i} \mathbf{S}^{(i)}\boldsymbol{\Omega}(i, k)\mathbf{W}^{(k)}\underline{\mathbf{b}}^{(k-1)}$$
$$+ \mathbf{S}^{(i)}\boldsymbol{\Omega}(i, m+1)\mathbf{W}^{(m+1)}\mathbf{D}^{(m)}\mathbf{b}^{(m)} + \mathbf{S}^{(i)}\boldsymbol{\Omega}(i, m+1)\mathbf{W}^{(m+1)}\underline{\mathbf{b}}^{(k)}$$

$$= \sum_{k=m+1}^{i} \mathbf{S}^{(i)}\boldsymbol{\Omega}(i, k)\mathbf{b}^{(k)} + \sum_{k=m+2}^{i} \mathbf{S}^{(i)}\boldsymbol{\Omega}(i, k)\mathbf{W}^{(k)}\underline{\mathbf{b}}^{(k-1)} + \mathbf{S}^{(i)}\boldsymbol{\Omega}(i, m)\mathbf{b}^{(m)}$$
$$+ \mathbf{S}^{(i)}\boldsymbol{\Omega}(i, m+1)\mathbf{W}^{(m+1)}\underline{\mathbf{b}}^{(m)}$$

$$= \sum_{k=m}^{i} \mathbf{S}^{(i)}\boldsymbol{\Omega}(i, k)\mathbf{b}^{(k)} + \sum_{k=m+1}^{i} \mathbf{S}^{(i)}\boldsymbol{\Omega}(i, k)\mathbf{W}^{(k)}\underline{\mathbf{b}}^{(k-1)}$$

$$= \mathbf{q}_i^{(m-1)}$$

Thus, $\boldsymbol{a}' = \boldsymbol{a}^{(m-1)}$, $\mathbf{P}' = \mathbf{P}^{(m-1)}$, $\mathbf{q}' = \mathbf{q}^{(m-1)}$ and $c' = c^{(m-1)}$ so the induction holds for layer $\hat{z}^{(m-1)}$:

$$\min_{\substack{x \in \mathcal{C} \\ z \in \mathcal{Z}}} f(x) \ge \max_{\tilde{\boldsymbol{\beta}}^{(m)} \ge 0} \min_{\substack{x \in \mathcal{C} \\ \tilde{z}^{(m-1)} \in \tilde{\mathcal{Z}}^{(m-1)}}} (\boldsymbol{a}^{(m-1)} + \mathbf{P}^{(m-1)}\tilde{\boldsymbol{\beta}}^{(m)})^{\top}\hat{z}^{(m-1)} + \mathbf{q}^{(m-1)\top}\tilde{\boldsymbol{\beta}}^{(m)} + c^{(m-1)}$$

$$(17)$$

Finally, Theorem 3.1 becomes the special case where $m = 0$ in Eq. 11, Eq. 12, Eq. 13 and Eq. 14. $\quad\square$

The next Lemma unveils the connection with CROWN [46] and is also useful for drawing connections to the dual problem.

**Lemma A.2.** *With $\mathbf{D}$, $\underline{\mathbf{b}}$ and $\mathbf{A}$ defined in Eq. 8, Eq. 9 and Eq. 10, we can rewrite Eq. 3 in Theorem 3.1 as:*

$$\min_{\substack{x \in \mathcal{C} \\ z \in \mathcal{Z}}} f(x) \geq \max_{\boldsymbol{\beta} \geq 0} \min_{x \in \mathcal{C}} \mathbf{A}^{(0)}x + \sum_{i=1}^{L-1} \mathbf{A}^{(i)}(\mathbf{D}^{(i)}\mathbf{b}^{(i)} + \underline{\mathbf{b}}^{(i)}) \tag{18}$$

*where $\mathbf{A}^{(i)}$, $0 \leq i \leq L-1$ contains variables $\boldsymbol{\beta}$.*

*Proof.* To prove this lemma, we simply follow the definition of $\mathbf{A}^{(i)}$ and check the resulting terms are the same as Eq. 3. For example,

$$\begin{aligned}
\mathbf{A}^{(0)} &= (\mathbf{A}^{(1)}\mathbf{D}^{(1)} + \boldsymbol{\beta}^{(1)\top}\mathbf{S}^{(1)})\mathbf{W}^{(1)} \\
&= \mathbf{A}^{(1)}\mathbf{D}^{(1)}\mathbf{W}^{(1)} + \boldsymbol{\beta}^{(1)\top}\mathbf{S}^{(1)}\mathbf{W}^{(1)} \\
&= (\mathbf{A}^{(2)}\mathbf{D}^{(2)} + \boldsymbol{\beta}^{(2)\top}\mathbf{S}^{(2)})\mathbf{W}^{(2)}\mathbf{D}^{(1)}\mathbf{W}^{(1)} + \boldsymbol{\beta}^{(1)\top}\mathbf{S}^{(1)}\mathbf{W}^{(1)} \\
&= \mathbf{A}^{(2)}\mathbf{D}^{(2)}\mathbf{W}^{(2)}\mathbf{D}^{(1)}\mathbf{W}^{(1)} + \boldsymbol{\beta}^{(2)\top}\mathbf{S}^{(2)}\mathbf{W}^{(2)}\mathbf{D}^{(1)}\mathbf{W}^{(1)} + \boldsymbol{\beta}^{(1)\top}\mathbf{S}^{(1)}\mathbf{W}^{(1)} \\
&= \cdots \\
&= \boldsymbol{\Omega}(L, 1)\mathbf{W}^{(1)} + \sum_{i=1}^{L-1} \boldsymbol{\beta}^{(i)\top}\mathbf{S}^{(i)}\boldsymbol{\Omega}(i, 1)\mathbf{W}^{(1)} \\
&= [\boldsymbol{a} + \mathbf{P}\boldsymbol{\beta}]^\top
\end{aligned}$$

Other terms can be shown similarly.

$\square$

With this definition of $\mathbf{A}$, we can see Eq. 3 as a modified form of CROWN, with an extra term $\boldsymbol{\beta}^{(i+1)\top}\mathbf{S}^{(i+1)}$ added when computing $\mathbf{A}^{(i)}$. When we set $\boldsymbol{\beta} = 0$, we obtain the same bound propagation rule for $\mathbf{A}$ as in CROWN. Thus, only a small change is needed to implement $\beta$-CROWN given an existing CROWN implementation: we add $\boldsymbol{\beta}^{(i+1)\top}\mathbf{S}^{(i+1)}$ after the linear bound propagates backwards through a ReLU layer. We also have the same observation in the dual space, as we will show this connection in the next subsection.

## A.2 Proofs for the connection to the dual space

**Theorem 3.2.** *The objective $d_{LP}$ for the dual problem of Eq. 10 can be represented as*

$$d_{LP} = -\|\boldsymbol{a} + \mathbf{P}\boldsymbol{\beta}\|_1 \cdot \epsilon + (\mathbf{P}^\top x_0 + \mathbf{q})^\top \boldsymbol{\beta} + \boldsymbol{a}^\top x_0 + c,$$

*where $\boldsymbol{a}$, $\mathbf{P}$, $\mathbf{q}$ and $c$ are defined in the same way as in Theorem 3.1, and $\boldsymbol{\beta} \geq 0$ corresponds to the dual variables of neuron split constraints in Eq. 10.*

*Proof.* To prove the Theorem 3.2, we demonstrate the detailed dual objective $d_{\text{LP}}$ for Eq. 10, following a construction similar to the one in Wong and Kolter [42]. We first associate a dual variable for each constraint involved in Eq. 10 including dual variables $\boldsymbol{\beta}$ for the per-neuron split constraints introduced by BaB. Although it is possible to directly write the dual LP for Eq. 10, for easier understanding, we

first rewrite the original primal verification problem into its Lagrangian dual form as Eq. 19, with dual variables $\boldsymbol{\nu}, \boldsymbol{\xi}^+, \boldsymbol{\xi}^-\boldsymbol{\mu}, \boldsymbol{\gamma}, \boldsymbol{\lambda}, \boldsymbol{\beta}$:

$$
\begin{aligned}
L(z, \hat{z}; \boldsymbol{\nu}, \boldsymbol{\xi}, \boldsymbol{\mu}, \boldsymbol{\gamma}, \boldsymbol{\lambda}, \boldsymbol{\beta}) = {} & z^{(L)} + \sum_{i=1}^{L} \boldsymbol{\nu}^{(i)\top}(z^{(i)} - \mathbf{W}^{(i)}\hat{z}^{(i-1)} - \mathbf{b}^{(i)}) \\
& + \boldsymbol{\xi}^{+\top}(\hat{z}^{(0)} - x_0 - \epsilon) + \boldsymbol{\xi}^{-\top}(-\hat{z}^{(0)} + x_0 - \epsilon) \\
& + \sum_{i=1}^{L-1} \sum_{\substack{j \notin \mathcal{Z}^{+(i)} \bigcup \mathcal{Z}^{-(i)} \\ \mathbf{l}_j^{(i)} < 0 < \mathbf{u}_j^{(i)}}} \left[ \boldsymbol{\mu}_j^{(i)\top}(-\hat{z}_j^{(i)}) + \boldsymbol{\gamma}_j^{(i)\top}(z_j^{(i)} - \hat{z}_j^{(i)}) + \boldsymbol{\lambda}_j^{(i)\top}(-\mathbf{u}_j^{(i)}z_j^{(i)} + (\mathbf{u}_j^{(i)} - \mathbf{l}_j^{(i)})\hat{z}_j^{(i)} + \mathbf{u}_j^{(i)}\mathbf{l}_j^{(i)}) \right] \\
& + \sum_{i=1}^{L-1} \left[ \sum_{z_j^{(i)} \in \mathcal{Z}^{-(i)}} \boldsymbol{\beta}_j^{(i)}z_j^{(i)} + \sum_{z_j^{(i)} \in \mathcal{Z}^{+(i)}} -\boldsymbol{\beta}_j^{(i)}z_j^{(i)} \right]
\end{aligned}
$$

Subject to:

$$
\boldsymbol{\xi}^+ \geq 0, \boldsymbol{\xi}^- \geq 0, \boldsymbol{\mu} \geq 0, \boldsymbol{\gamma} \geq 0, \boldsymbol{\lambda} \geq 0, \boldsymbol{\beta} \geq 0 \tag{19}
$$

The original minimization problem then becomes:

$$
\max_{\boldsymbol{\nu}, \boldsymbol{\xi}^+, \boldsymbol{\xi}^-, \boldsymbol{\mu}, \boldsymbol{\gamma}, \boldsymbol{\lambda}, \boldsymbol{\beta}} \min_{z, \hat{z}} L(z, \hat{z}, \boldsymbol{\nu}, \boldsymbol{\xi}^+, \boldsymbol{\xi}^-, \boldsymbol{\mu}, \boldsymbol{\gamma}, \boldsymbol{\lambda}, \boldsymbol{\beta})
$$

Given fixed intermediate bounds $\mathbf{l}, \mathbf{u}$, the inner minimization is a linear optimization problem and we can simply transfer it to the dual form. To further simplify the formula, we introduce notations similar to those in [42], where $\hat{\boldsymbol{\nu}}^{(i-1)} = \mathbf{W}^{(i)\top}\boldsymbol{\nu}^{(i)}$ and $\boldsymbol{\alpha}_j^{(i)} = \frac{\boldsymbol{\gamma}_j^{(i)}}{\boldsymbol{\mu}_j^{(i)} + \boldsymbol{\gamma}_j^{(i)}}$. Then the dual form can be written as Eq. 20.

$$
\max_{0 \leq \boldsymbol{\alpha} \leq 1, \boldsymbol{\beta} \geq 0} g(\boldsymbol{\alpha}, \boldsymbol{\beta}), \text{ where}
$$

$$
g(\boldsymbol{\alpha}, \boldsymbol{\beta}) = -\sum_{i=1}^{L} \boldsymbol{\nu}^{(i)\top}\mathbf{b}^{(i)} - \hat{\boldsymbol{\nu}}^{(0)\top}x_0 - ||\hat{\boldsymbol{\nu}}^{(0)}||_1 \cdot \epsilon + \sum_{i=1}^{L-1} \sum_{\substack{j \notin \mathcal{Z}^{+(i)} \bigcup \mathcal{Z}^{-(i)} \\ \mathbf{l}_j^{(i)} < 0 < \mathbf{u}_j^{(i)}}} \mathbf{l}_j^{(i)}[\boldsymbol{\nu}_j^{(i)}]^+
$$

Subject to:

$$
\begin{aligned}
& \boldsymbol{\nu}^{(L)} = -1, \hat{\boldsymbol{\nu}}^{(i-1)} = \mathbf{W}^{(i)\top}\boldsymbol{\nu}^{(i)}, \quad i \in \{1, \ldots, L\} \\
& \boldsymbol{\nu}_j^{(i)} = 0, \quad \text{when } \mathbf{u}_j^{(i)} \leq 0, i \in \{1, \ldots, L-1\} \\
& \boldsymbol{\nu}_j^{(i)} = \hat{\boldsymbol{\nu}}_j^{(i)}, \quad \text{when } \mathbf{l}_j^{(i)} \geq 0, i \in \{1, \ldots, L-1\} \\
& \left. \begin{aligned} & [\boldsymbol{\nu}_j^{(i)}]^+ = \mathbf{u}_j^{(i)}\boldsymbol{\lambda}_j^{(i)}, [\boldsymbol{\nu}_j^{(i)}]^- = \boldsymbol{\alpha}_j^{(i)}[\hat{\boldsymbol{\nu}}_j^{(i)}]^- \\ & \boldsymbol{\lambda}_j^{(i)} = \frac{[\hat{\boldsymbol{\nu}}_j^{(i)}]^+}{\mathbf{u}_j^{(i)} - \mathbf{l}_j^{(i)}}, \boldsymbol{\alpha}_j^{(i)} = \frac{\boldsymbol{\gamma}_j^{(i)}}{\boldsymbol{\mu}_j^{(i)} + \boldsymbol{\gamma}_j^{(i)}} \end{aligned} \right\} \text{ when } \mathbf{l}_j^{(i)} < 0 < \mathbf{u}_j^{(i)}, j \notin \mathcal{Z}^{+(i)} \bigcup \mathcal{Z}^{-(i)}, i \in \{1, \ldots, L-1\} \\
& \boldsymbol{\nu}_j^{(i)} = -\boldsymbol{\beta}_j^{(i)}, \quad j \in \mathcal{Z}^{-(i)}, i \in \{1, \ldots, L-1\} \\
& \boldsymbol{\nu}_j^{(i)} = \boldsymbol{\beta}_j^{(i)} + \hat{\boldsymbol{\nu}}_j^{(i)}, \quad j \in \mathcal{Z}^{+(i)}, i \in \{1, \ldots, L-1\} \\
& \boldsymbol{\mu} \geq 0, \boldsymbol{\gamma} \geq 0, \boldsymbol{\lambda} \geq 0, \boldsymbol{\beta} \geq 0, 0 \leq \boldsymbol{\alpha} \leq 1
\end{aligned} \tag{20}
$$

Similar to the dual form in [42] (our differences are highlighted in blue), the dual problem can be viewed in the form of another deep network by backward propagating $\boldsymbol{\nu}^{(L)}$ to $\hat{\boldsymbol{\nu}}^{(0)}$ following the rules in Eq. 20. If we look closely at the conditions and coefficients when backward propagating $\boldsymbol{\nu}_j^{(i)}$ for $j$-th ReLU at layer $i$ in Eq. 20, we can observe that they match exactly to the propagation of

diagonal matrices $\mathbf{D}^{(i)}, \mathbf{S}^{(i)}$, and vector $\underline{\mathbf{b}}^{(i)}$ defined in Eq. 8 and Eq. 9. Therefore, using notations in Eq. 8 and Eq. 9 we can essentially simplify the dual LP problem in Eq. 20 to:

$$\boldsymbol{\nu}^{(L)} = -1, \hat{\boldsymbol{\nu}}^{(i-1)} = \mathbf{W}^{(i)\top}\boldsymbol{\nu}^{(i)}, \boldsymbol{\nu}^{(i)} = \mathbf{D}^{(i)}\hat{\boldsymbol{\nu}}^{(i)} - \boldsymbol{\beta}^{(i)}\mathbf{S}^{(i)}, i \in \{L, \cdots, 1\}$$

$$\sum_{\substack{\mathbf{l}_j^{(i)} < 0 < \mathbf{u}_j^{(i)} \\ j \notin \mathcal{Z}^{+(i)} \cup \mathcal{Z}^{-(i)}}} \mathbf{l}_j^{(i)}[\boldsymbol{\nu}_j^{(i)}]^+ = -\hat{\boldsymbol{\nu}}^{(i)T}\underline{\mathbf{b}}^{(i)}, j \in \{1, \cdots, d_i\}, i \in \{L-1, \cdots, 1\} \tag{21}$$

Then we prove the key claim for this proof with induction where $\boldsymbol{a}^{(m)}$ and $\mathbf{P}^{(m)}$ are defined in Eq. 11 and Eq. 12:

$$\hat{\boldsymbol{\nu}}^{(m)} = -\boldsymbol{a}^{(m)} - \mathbf{P}^{(m)}\tilde{\boldsymbol{\beta}}^{(m+1)} \tag{22}$$

When $m = L - 1$, we can have $\hat{\boldsymbol{\nu}}^{(L-1)} = -\boldsymbol{a}^{(L-1)} - \mathbf{P}^{(L-1)}\tilde{\boldsymbol{\beta}}^{(L)} = -\left[\boldsymbol{\Omega}(L,L)\mathbf{W}^{(L)}\right]^\top - \mathbf{0} = -\mathbf{W}^{(L)\top}$ which is true according to Eq. 21.

Now we assume that $\hat{\boldsymbol{\nu}}^{(m)} = -\boldsymbol{a}^{(m)} - \mathbf{P}^{(m)}\tilde{\boldsymbol{\beta}}^{(m+1)}$ holds, and we show that $\hat{\boldsymbol{\nu}}^{(m-1)} = -\boldsymbol{a}^{(m-1)} - \mathbf{P}^{(m-1)}\tilde{\boldsymbol{\beta}}^{(m)}$ will hold as well:

$$\hat{\boldsymbol{\nu}}^{(m-1)} = \mathbf{W}^{(m)\top}\left(\mathbf{D}^{(m)}\hat{\boldsymbol{\nu}}^{(m)} - \boldsymbol{\beta}^{(m)}\mathbf{S}^{(m)}\right)$$

$$= -\mathbf{W}^{(m)\top}\mathbf{D}^{(m)}\boldsymbol{a}^{(m)} - \mathbf{W}^{(m)\top}\mathbf{D}^{(m)}\mathbf{P}^{(m)}\tilde{\boldsymbol{\beta}}^{(m+1)} - \mathbf{W}^{(m)\top}\boldsymbol{\beta}^{(m)}\mathbf{S}^{(m)}$$

$$= -\boldsymbol{a}^{(m-1)} - \left[\left(\mathbf{S}^{(m)}\mathbf{W}^{(m)}\right)^\top, \left(\mathbf{P}^{(m)\top}\mathbf{D}^{(m)}\mathbf{W}^{(m)}\right)^\top\right]\left[\boldsymbol{\beta}^{(m)}, \tilde{\boldsymbol{\beta}}^{(m+1)}\right]$$

$$= -\boldsymbol{a}^{(m-1)} - \mathbf{P}^{(m-1)}\tilde{\boldsymbol{\beta}}^{(m)}$$

Therefore, the claim Eq. 22 is proved with induction. Lastly, we prove the following claim where $\mathbf{q}^{(m)}$ and $c^{(m)}$ are defined in Eq. 13 and Eq. 14.

$$-\sum_{i=m+1}^{L} \boldsymbol{\nu}^{(i)\top}\mathbf{b}^{(i)} + \sum_{i=m+1}^{L-1} \sum_{\substack{\mathbf{l}_j^{(i)} < 0 < \mathbf{u}_j^{(i)} \\ j \notin \mathcal{Z}^{+(i)} \cup \mathcal{Z}^{-(i)}}} \mathbf{l}_j^{(i)}[\boldsymbol{\nu}_j^{(i)}]^+ = \mathbf{q}^{(m)\top}\tilde{\boldsymbol{\beta}}^{(m+1)} + c^{(m)} \tag{23}$$

This claim can be proved by applying Eq. 21 and Eq. 22.

$$-\sum_{i=m+1}^{L} \boldsymbol{\nu}^{(i)\top}\mathbf{b}^{(i)} + \sum_{i=m+1}^{L-1} \sum_{\substack{\mathbf{l}_j^{(i)} < 0 < \mathbf{u}_j^{(i)} \\ j \notin \mathcal{Z}^{+(i)} \cup \mathcal{Z}^{-(i)}}} \mathbf{l}_j^{(i)}[\boldsymbol{\nu}_j^{(i)}]^+$$

$$= -\sum_{i=m+1}^{L} \left(\mathbf{D}^{(i)}\hat{\boldsymbol{\nu}}^{(i)} - \boldsymbol{\beta}^{(i)}\mathbf{S}^{(i)}\right)^\top \mathbf{b}^{(i)} + \sum_{i=m+2}^{L} \left(-\hat{\boldsymbol{\nu}}^{(i-1)T}\underline{\mathbf{b}}^{(i-1)}\right)$$

$$= \sum_{i=m+1}^{L} \left[\left(\boldsymbol{a}^{(i)\top} + \tilde{\boldsymbol{\beta}}^{(i+1)\top}\mathbf{P}^{(i)\top}\right)\mathbf{D}^{(i)}\mathbf{b}^{(i)} + \boldsymbol{\beta}^{(i)\top}\mathbf{S}^{(i)}\mathbf{b}^{(i)}\right]$$

$$+ \sum_{i=m+2}^{L} \left(\boldsymbol{a}^{(i-1)\top} + \tilde{\boldsymbol{\beta}}^{(i)\top}\mathbf{P}^{(i-1)\top}\right)\underline{\mathbf{b}}^{(i-1)}$$

$$= \sum_{i=m+1}^{L} \tilde{\boldsymbol{\beta}}^{(i)\top}\left[\mathbf{S}^{(i)}, \mathbf{P}^{(i)\top}\mathbf{D}^{(i)}\right]\mathbf{b}^{(i)} + \sum_{i=m+2}^{L} \tilde{\boldsymbol{\beta}}^{(i)\top}\mathbf{P}^{(i-1)\top}\underline{\mathbf{b}}^{(i-1)}$$

$$+ \sum_{i=m+1}^{L} \boldsymbol{a}^{(i)\top}\mathbf{D}^{(i)}\mathbf{b}^{(i)} + \sum_{i=m+2}^{L} \boldsymbol{a}^{(i-1)\top}\underline{\mathbf{b}}^{(i-1)}$$

$$= \mathbf{q}^{(m)\top}\tilde{\boldsymbol{\beta}}^{(m+1)} + c^{(m)}$$

Finally, we apply claims Eq. 22 and Eq. 23 into the dual form solution Eq. 20 and prove the Theorem 3.2.

$$g(\boldsymbol{\alpha}, \boldsymbol{\beta}) = -\sum_{i=1}^{L} \boldsymbol{\nu}^{(i)\top} \mathbf{b}^{(i)} - \hat{\boldsymbol{\nu}}^{(0)\top} x_0 - ||\hat{\boldsymbol{\nu}}^{(0)}||_1 \cdot \epsilon + \sum_{i=1}^{L-1} \sum_{\substack{\mathbf{l}_j^{(i)} < 0 < \mathbf{u}_j^{(i)} \\ j \notin \mathcal{Z}^{+(i)} \bigcup \mathcal{Z}^{-(i)}}} \mathbf{l}_j^{(i)}[\boldsymbol{\nu}_j^{(i)}]^+$$

$$= -||-\boldsymbol{a}^{(0)} - \mathbf{P}^{(0)} \tilde{\boldsymbol{\beta}}^{(1)}||_1 \cdot \epsilon + \left( \boldsymbol{a}^{(0)\top} + \tilde{\boldsymbol{\beta}}^{(1)\top} \mathbf{P}^{(0)\top} \right) x_0 + \mathbf{q}^{(0)\top} \tilde{\boldsymbol{\beta}}^{(1)} + c^{(0)}$$

$$= -||\boldsymbol{a} + \mathbf{P} \tilde{\boldsymbol{\beta}}^{(1)}||_1 \cdot \epsilon + \left( \mathbf{P}^\top x_0 + \mathbf{q} \right)^\top \tilde{\boldsymbol{\beta}}^{(1)} + \boldsymbol{a}^\top x_0 + c$$

**A more intuitive proof.** Here we provide another intuitive proof showing why the dual form solution of verification objective in Eq. 20 is the same as the primal one in Thereom 3.1. $d_{\text{LP}} = g(\boldsymbol{\alpha}, \boldsymbol{\beta})$ is the dual objective for Eq. 10 with free variables $\boldsymbol{\alpha}$ and $\boldsymbol{\beta}$. We want to show that the dual problem can be viewed in the form of backward propagating $\boldsymbol{\nu}^{(L)}$ to $\hat{\boldsymbol{\nu}}^{(0)}$ following the same rules in $\beta$-CROWN. Salman et al. [30] showed that CROWN computes the same solution as the dual form in Wong and Kolter [42]: $\hat{\boldsymbol{\nu}}^{(i)}$ is corresponding to $-\mathbf{A}^{(i)}$ in CROWN (defined in the same way as in Eq. 10 but with $\boldsymbol{\beta}^{(i+1)} = 0$) and $\boldsymbol{\nu}^{(i)}$ is corresponding to $-\mathbf{A}^{(i+1)} \mathbf{D}^{(i+1)}$. When the split constraints are introduced, extra terms for the dual variable $\boldsymbol{\beta}$ modify $\boldsymbol{\nu}^{(i)}$ (highlighted in blue in Eq. 20). The way $\beta$-CROWN modifies $\mathbf{A}^{(i+1)} \mathbf{D}^{(i+1)}$ is exactly the same as the way $\boldsymbol{\beta}^{(i)}$ affects $\boldsymbol{\nu}^{(i)}$: when we split $z_j^{(i)} \geq 0$, we add $\boldsymbol{\beta}_j^{(i)}$ to the $\boldsymbol{\nu}_j^{(i)}$ in Wong and Kolter [42]; when we split $z_j^{(i)} \geq 0$, we add $-\boldsymbol{\beta}_j^{(i)}$ to the $\boldsymbol{\nu}_j^{(i)}$ in Wong and Kolter [42] ($\boldsymbol{\nu}_j^{(i)}$ is 0 in this case because it is set to be inactive). To make this relationship more clear, we define a new variable $\boldsymbol{\nu}'$, and rewrite relevant terms involving $\boldsymbol{\nu}, \hat{\boldsymbol{\nu}}$ below:

$$\begin{aligned} \boldsymbol{\nu}_j^{(i)} &= 0, \quad j \in \mathcal{Z}^{-(i)}; \\ \boldsymbol{\nu}_j^{(i)} &= \hat{\boldsymbol{\nu}}_j^{(i)}, \quad j \in \mathcal{Z}^{+(i)}; \\ \boldsymbol{\nu}_j^{(i)} &\text{ is defined in the same way as in Eq. 20 for other cases} \\ \boldsymbol{\nu}_j^{(i)\prime} &= -\boldsymbol{\beta}_j^{(i)} + \boldsymbol{\nu}_j^{(i)}, \quad j \in \mathcal{Z}^{-(i)}; \\ \boldsymbol{\nu}_j^{(i)\prime} &= \boldsymbol{\beta}_j^{(i)} + \boldsymbol{\nu}_j^{(i)}, \quad j \in \mathcal{Z}^{+(i)}; \\ \boldsymbol{\nu}_j^{(i)\prime} &= \boldsymbol{\nu}_j^{(i)}, \quad \text{otherwise} \\ \hat{\boldsymbol{\nu}}^{(i-1)} &= \mathbf{W}^{(i)\top} \boldsymbol{\nu}^{(i)\prime}; \end{aligned} \quad (24)$$

It is clear that $\boldsymbol{\nu}'$ corresponds to the term $-(\mathbf{A}^{(i+1)} \mathbf{D}^{(i+1)} + \boldsymbol{\beta}^{(i+1)\top} \mathbf{S}^{(i+1)})$ in Eq. 10, by noting that $\boldsymbol{\nu}^{(i)}$ in [42] is equivalent to $-\mathbf{A}^{(i+1)} \mathbf{D}^{(i+1)}$ in CROWN and the choice of signs in $\mathbf{S}^{(i+1)}$ reflects neuron split constraints. Thus, the dual formulation will produce the same results as Eq. 18, and thus also equivalent to Eq. 3.

$\square$

**Corollary 3.2.1.** *When $\boldsymbol{\alpha}$ and $\boldsymbol{\beta}$ are optimally set, $\beta$-CROWN produces the same solution as LP with split constraints when intermediate bounds $\mathbf{l}, \mathbf{u}$ are fixed. Formally,*

$$\max_{0 \leq \boldsymbol{\alpha} \leq 1, \boldsymbol{\beta} \geq 0} g(\boldsymbol{\alpha}, \boldsymbol{\beta}) = p_{LP}^*$$

*where $p_{LP}^*$ is the optimal objective of Eq. 10.*

*Proof.* Given fixed intermediate layer bounds $\mathbf{l}$ and $\mathbf{u}$, the dual form of the verification problem in Eq. 10 is a linear programming problem with dual variables defined in Eq. 19. Suppose we use an LP solver to obtain the optimal dual solution $\boldsymbol{\nu}^*, \boldsymbol{\xi}^*, \boldsymbol{\mu}^*, \boldsymbol{\gamma}^*, \boldsymbol{\lambda}^*, \boldsymbol{\beta}^*$. Then we can set $\alpha_j^{(i)} = \frac{\gamma_j^{(i)*}}{\mu_j^{(i)*} + \gamma_j^{(i)*}}$, $\boldsymbol{\beta} = \boldsymbol{\beta}^*$ and plug them into Eq. 20 to get the optimal dual solution $d_{\text{LP}}^*$. Theorem 3.2 shows that, $\beta$-CROWN can compute the same objective $d_{\text{LP}}^*$ given the same $\alpha_j^{(i)} = \frac{\gamma_j^{(i)*}}{\mu_j^{(i)*} + \gamma_j^{(i)*}}$, $\boldsymbol{\beta} = \boldsymbol{\beta}^*$,

thus $\max_{0 \leq \boldsymbol{\alpha} \leq 1, \boldsymbol{\beta} \geq 0} g(\boldsymbol{\alpha}, \boldsymbol{\beta}) \geq d_{\mathrm{LP}}^*$. On the other hand, for any setting of $\boldsymbol{\alpha}$ and $\boldsymbol{\beta}$, $\beta$-CROWN produces the same solution $g(\boldsymbol{\alpha}, \boldsymbol{\beta})$ as the rewritten dual LP in Eq. 20, so $g(\boldsymbol{\alpha}, \boldsymbol{\beta}) \leq d_{\mathrm{LP}}^*$. Thus, we have $\max_{0 \leq \boldsymbol{\alpha} \leq 1, \boldsymbol{\beta} \geq 0} g(\boldsymbol{\alpha}, \boldsymbol{\beta}) = d_{\mathrm{LP}}^*$. Finally, due to the strong duality in linear programming, $p_{\mathrm{LP}}^* = d_{\mathrm{LP}}^* = \max_{0 \leq \boldsymbol{\alpha} \leq 1, \boldsymbol{\beta} \geq 0} g(\boldsymbol{\alpha}, \boldsymbol{\beta})$. □

The variables $\boldsymbol{\alpha}$ in $\beta$-CROWN can be translated to dual variables in LP as well. Given $\boldsymbol{\alpha}$ in $\beta$-CROWN, we can get the corresponding dual LP variables $\boldsymbol{\mu}, \boldsymbol{\gamma}$ given $\boldsymbol{\alpha}$ by setting $\boldsymbol{\mu}_j^{(i)} = (1 - \boldsymbol{\alpha}_j^{(i)})[\hat{\boldsymbol{\nu}}_j^{(i)}]^-$ and $\boldsymbol{\gamma}_j^{(i)} = \boldsymbol{\alpha}_j^{(i)}[\hat{\boldsymbol{\nu}}_j^{(i)}]^-$.

### A.3 Proof for soundness and completeness

**Theorem 3.3.** *$\beta$-CROWN with branch and bound on splitting ReLU neurons is sound and complete.*

*Proof.* **Soundness.** Branch and bound (BaB) with $\beta$-CROWN is sound because for each subdomain $\mathcal{C}_i := \{x \in \mathcal{C}, z \in \mathcal{Z}_i\}$, we apply Theorem 3.1 to obtain a sound lower bound $\underline{f}_{\mathcal{C}_i}$ (the bound is valid for any $\boldsymbol{\beta} \geq 0$). The final bound returned by BaB is $\min_i \underline{f}_{\mathcal{C}_i}$ which represents the worst case over all subdomains, and is a sound lower bound for $x \in \mathcal{C} := \cup_i \mathcal{C}_i$.

**Completeness.** To show completeness, we need to solve Eq. 1 to its global minimum. When there are $N$ unstable neurons, we have up to $2^N$ subdomains, and in each subdomain we have all unstable ReLU neurons split into one of the $z_j^{(i)} \geq 0$ or $z_j^{(i)} < 0$ case. The final solution obtained by BaB is the min over these $2^N$ subdomains. To obtain the global minimum, we must ensure that in every of these $2^N$ subdomain we can solve Eq. 10 exactly.

When all unstable neurons are split in a subdomain $\mathcal{C}_i$, the network becomes a linear network and neuron split constraints become linear constraints w.r.t. inputs. Under this case, an LP with Eq. 10 can solve the verification problem in $\mathcal{C}_i$ exactly. In $\beta$-CROWN, we solve the subdomain using the usually non-concave formulation Eq. 12; however, in this case, it becomes concave in $\hat{\boldsymbol{\beta}}$ because no intermediate layer bounds are used (no $\boldsymbol{\alpha}'$ and $\boldsymbol{\beta}'$) and no ReLU neuron is relaxed (no $\boldsymbol{\alpha}$), thus the only optimizable variable is $\boldsymbol{\beta}$ (Eq. 12 becomes Eq. 8). Eq. 8 is concave in $\boldsymbol{\beta}$ so (super)gradient ascent guarantees to converge to the global optimal $\boldsymbol{\beta}^*$. To ensure convergence without relying on a preset learning rate, a line search can be performed in this case. Then, according to Corollary 3.2.1, this optimal $\boldsymbol{\beta}^*$ corresponds to the optimal dual variable for the LP in Eq. 10 and the objective is a global minimum of Eq. 10. □

## B  More details on $\beta$-CROWN with branch and bound (BaB)

### B.1  $\beta$-CROWN with branch and bound for complete verification

We list our $\beta$-CROWN with branch and bound based complete verifier ($\beta$-CROWN BAB) in Algorithm 1. The algorithm takes a target NN function $f$ and a domain $\mathcal{C}$ as inputs. The subprocedure `optimized_beta_CROWN` optimizes $\hat{\boldsymbol{\alpha}}$ and $\hat{\boldsymbol{\beta}}$ (free variables for computing intermediate layer bounds and last layer bounds) as Eq. 12 in Section 3.3. It operates in a batch and returns the lower and upper bounds for $n$ selected subdomains simultaneously: a lower bound is obtained by optimizing Eq. 12 using $\beta$-CROWN and an upper bound can be the network prediction $f(x^*)$ given the $x^*$ that minimizes Eq. 7[1]. Initially, we don't have any splits, so we only need to optimize $\hat{\boldsymbol{\alpha}}$ to obtain $\underline{f}$ for $x \in \mathcal{C}$ (Line 2). Then we utilize the power of GPUs to split in parallel and maintain a global set $\mathbb{P}$ storing all the sub-domains which does not satisfy $\underline{f}_{\mathcal{C}_i} < 0$ (Line 5-10). Specifically, `batch_pick_out` extends branching strategy BaBSR [8] or FSB [11] in a parallel manner to select $n$ (batch size) sub-domains in $\mathbb{P}$ and determine the corresponding ReLU neuron to split for each of them. If the length of $\mathbb{P}$ is less than $n$, then we reduce $n$ to the length of $\mathbb{P}$. `batch_split` splits each selected $\mathcal{C}_i$ to two sub-domains $\mathcal{C}_i^l$ and $\mathcal{C}_i^u$ by forcing the selected unstable ReLU neuron to be positive and negative, respectively. `Domain_Filter` filters out verified sub-domains (proved with $\underline{f}_{\mathcal{C}_i} \geq 0$) and we insert the remaining

---

[1]We want an upper bound of the objective in Eq. 1. Since Eq. 1 is an minimization problem, any feasible $x$ produces an upper bound of the optimal objective. When Eq. 1 is solved exactly as $f^*$ (such as in the case where all neurons are split), we have $f^* = \underline{f} = \overline{f}$. See also the discussions in Section I.1 of De Palma et al. [10].

**Algorithm 1** $\beta$-CROWN with branch and bound for complete verification. Comments are in brown.

---

1: **Inputs**: $f, \mathcal{C}, n$ (batch size), $\delta$ (tolerance), $\eta$ (maximum length of sub-domains)
2: $(\underline{f}, \overline{f}) \leftarrow \texttt{optimized\_beta\_CROWN}(f, [\mathcal{C}])$    ▷ Initially there is no split, so optimization is done over $\hat{\alpha}$
3: $\mathbb{P} \leftarrow [(\underline{f}, \overline{f}, \mathcal{C})]$    ▷ $\mathbb{P}$ is the set of all unverified sub-domains
4: **while** $\underline{f} < 0$ **and** $\overline{f} \geq 0$ **and** $\overline{f} - \underline{f} > \delta$ **and** $\texttt{length}(\mathbb{P}) < \eta$ **do**
5:    $(\mathcal{C}_1, \ldots, \mathcal{C}_n) \leftarrow \texttt{batch\_pick\_out}(\mathbb{P}, n)$    ▷ Pick sub-domains to split and removed them from $\mathbb{P}$
6:    $[\mathcal{C}_1^l, \mathcal{C}_1^u, \ldots, \mathcal{C}_n^l, \mathcal{C}_n^u] \leftarrow \texttt{batch\_split}(\mathcal{C}_1, \ldots, \mathcal{C}_n)$ ▷ Each $\mathcal{C}_i$ splits into two sub-domains $\mathcal{C}_i^l$ and $\mathcal{C}_i^u$
7:    $\left[\underline{f}_{\mathcal{C}_1^l}, \overline{f}_{\mathcal{C}_1^l}, \underline{f}_{\mathcal{C}_1^u}, \overline{f}_{\mathcal{C}_1^u}, \ldots, \underline{f}_{\mathcal{C}_n^l}, \overline{f}_{\mathcal{C}_n^l}, \underline{f}_{\mathcal{C}_n^u}, \overline{f}_{\mathcal{C}_n^u}\right] \leftarrow \texttt{optimized\_beta\_CROWN}(f, [\mathcal{C}_1^l, \mathcal{C}_1^u, \ldots, \mathcal{C}_n^l, \mathcal{C}_n^u])$
      ▷ Compute lower and upper bounds by optimizing $\hat{\alpha}$ and $\hat{\beta}$ mentioned in Section 3.3 in a batch
8:    $\mathbb{P} \leftarrow \mathbb{P} \bigcup \texttt{Domain\_Filter}\left([\underline{f}_{\mathcal{C}_1^l}, \overline{f}_{\mathcal{C}_1^l}, \mathcal{C}_1^l], [\underline{f}_{\mathcal{C}_1^u}, \overline{f}_{\mathcal{C}_1^u}, \mathcal{C}_1^u], \ldots, [\underline{f}_{\mathcal{C}_n^l}, \overline{f}_{\mathcal{C}_n^1}, \mathcal{C}_n^l], [\underline{f}_{\mathcal{C}_n^u}, \overline{f}_{\mathcal{C}_n^u}, \mathcal{C}_n^u]\right)$    ▷
      Filter out verified sub-domains, insert the left domains back to $\mathbb{P}$
9:    $\underline{f} \leftarrow \min\{\underline{f}_{\mathcal{C}_i} \mid (\underline{f}_{\mathcal{C}_i}, \overline{f}_{\mathcal{C}_i}, \mathcal{C}_i) \in \mathbb{P}\}, i = 1, \ldots, n$ ▷ To ease notation, $\mathcal{C}_i$ here indicates either $\mathcal{C}_i^u$ or $\mathcal{C}_i^l$
10:    $\overline{f} \leftarrow \min\{\overline{f}_{\mathcal{C}_i} \mid (\underline{f}_{\mathcal{C}_i}, \overline{f}_{\mathcal{C}_i}, \mathcal{C}_i) \in \mathbb{P}\}, i = 1, \ldots, n$
11: **Outputs:** $\underline{f}, \overline{f}$

---

ones to $\mathbb{P}$. The loop breaks if the property is proved ($\underline{f} \geq 0$), or a counter-example is found in any sub-domain ($\overline{f} < 0$), or the lower bound $\underline{f}$ and upper bound $\overline{f}$ are sufficiently close, or the length of sub-domains $\mathbb{P}$ reaches a desired threshold $\eta$ (maximum memory limit).

Note that for models evaluated in our paper, we find that computing intermediate layer bounds in every iteration at line 7 is too costly (although it is possible and supported) so we only compute intermediate layer bounds once at line 2. At line 7, only the neuron with split constraints have their intermediate layer bounds updated, and other intermediate bounds are not recomputed. This makes the intermediate layer bounds looser but it allows us to quickly explore a large number of nodes on the branch and bound search tree and is overall beneficial for verifying most models. A similar observation was also found in De Palma et al. [10] (Section 5.1.1).

## B.2   Comparisons to other GPU based complete verifiers

Bunel et al. [6] proposed to reformulate the linear programming problem in Eq. 10 through Lagrangian decomposition. Eq. 10 is decomposed layer by layer, and each layer is solved with simple closed form solutions on GPUs. A Lagrangian is used to enforce the equality between the output of a previous layer and the input of a later layer. This optimization formulation has the same power as a LP (Eq. 10) under convergence. The main drawback of this approach is that it converges relatively slowly (it typically requires hundreds of iterations to converge to a solution similar to the solution of a LP), and it also cannot easily jointly optimize intermediate layer bounds. In Table 1 (PROX BABSR) and Figure 1 (BDD+ BABSR, which refers to the same method) we can see that this approach is relatively slow and has high timeout rates compared to other GPU accelerated complete verifiers. Recently, De Palma et al. [11] proposed a better branching strategy, filtered smart branching (FSB), to further improved verification performance of [6], but the Lagrangian Decomposition based incomplete verifier and the branch and bound procedure stay the same.

De Palma et al. [10] used a tighter convex relaxation [2] than the typical LP formulation in Eq. 10 for the incomplete verifier. This tighter relaxation may contain exponentially many constraints, and De Palma et al. [10] proposed to solve the verification problem in its dual form where each constraint becomes a dual variable. A small active set of dual variables is maintained during dual optimization to ensure efficiency. This tighter relaxation allows it to outperform [6], but it also comes with extra computational costs and difficulties for an efficient implementation (e.g. a "masked" forward/backward pass is needed which requires a customised low-level convolution implementation). Additionally, De Palma et al. [10] did not optimize intermediate layer bounds jointly.

Xu et al. [45] used CROWN [46] (categorized as a linear relaxation based perturbation analysis (LiRPA) algorithm) as the incomplete solver in BaB. Since CROWN cannot encode neural split constraints, Xu et al. [45] essentially solve Eq. 10 *without* neuron split constraints ($z_j^{(i)} \geq 0, i \in \{1, \cdots, L-1\}, j \in \mathcal{Z}^{+(i)}$ and $z_j^{(i)} < 0, i \in \{1, \cdots, L-1\}, j \in \mathcal{Z}^{-(i)}$) in Eq. 10. The missing

constraints lead to looser bounds and unnecessary branches. Additionally, using CROWN as the incomplete solver leads to incompleteness - even when all unstable ReLU neurons are split, Xu et al. [45] still cannot solve Eq. 1 to a global minimum, so a LP solver has to be used to check inconsistent splits and guarantee completeness. Our $\beta$-CROWN BaB overcomes these drawbacks: we consider per-neuron split constraints in $\beta$-CROWN which reduces the number of branches and solving time (Table 1). Most importantly, $\beta$-CROWN with branch and bound is sound and complete (Theorem 3.3) and we do not rely on any LP solvers.

Another difference between Xu et al. [45] and our method is the joint optimization of intermediate layer bounds (Section 3.3). Although [45] also optimized intermediate layer bounds, they only optimize $\alpha$ and do not have $\beta$, and they share the same variable $\alpha$ for all intermediate layer bounds and final bounds, with a total of $O(Ld)$ variables to optimize. Our analysis in Section 3.3 shows that there are in fact, $O(L^2d^2)$ free variables to optimize, and we share less variables as in Xu et al. [45]. This allows us to achieve tighter bounds and improve overall performance.

### B.3  Detection of Infeasibility

Maximizing Eq. 8 with infeasible constraints leads to unbounded dual objective, which can be detected by checking if this optimized lower bound becomes *greater than the upper bound* (which is also maintained in BaB, see Alg.1 in Sec. B.1). For the robustness verification problem, a subdomain that has lower bound greater than 0 is dropped, which includes the unbounded case. Due to insufficient convergence, this cannot always detect infeasibility, but it *does not affect soundness*, as this infeasible subdomain only leads to worse overall lower bound in BaB. To guarantee *completeness*, we show that when all unstable neurons are split the problem is concave (see Section A.3); in this case, we can use line search to guarantee convergence when feasible, and detect infeasibility if the objective exceeds the upper bound (line search guarantees the objective can eventually exceed upper bound). In most real scenarios, the verifier either finishes or times out before all unstable neurons are split.

## C  Details on Experimental Setup and Results

### C.1  Experimental Setup

We run our experiments on a machine with a single NVIDIA RTX 3090 GPU (24GB GPU memory), a AMD Ryzen 9 5950X CPU and 64GB memory. Our $\beta$-CROWN solver uses 1 CPU and 1 GPU only, except for the MLP models in Table 2 where 16 threads are used to compute intermediate layer bounds with Gurobi[2]. We use the Adam optimizer [21] to solve both $\hat{\alpha}$ and $\hat{\beta}$ in Eq. 12 with 20 iterations. The learning rates are set as 0.1 and 0.05 for optimizing $\hat{\alpha}$ and $\hat{\beta}$ respectively. We decay the learning rates with a factor of 0.98 per iteration. To maximize the benefits of parallel computing on GPU, we use batch sizes $n = 1024$ for Base (CIFAR-10), Wide (CIFAR-10), Deep (CIFAR-10), CNN-A-Adv (MNIST) and ConvSmall (MNIST), $n = 2048$ for ConvSmall (CIFAR-10), $n = 4096$ for CNN-A-Adv (CIFAR-10), CNN-A-Adv-4 (CIFAR-10), CNN-A-Mix (CIFAR-10) and CNN-A-Mix-4 (CIFAR-10), $n = 256$ for CNN-B-Adv (CIFAR-10) and CNN-B-Adv-4 (CIFAR-10), $n = 1024$ for ConvBig (MNIST), $n = 10$ for ConvBig (CIFAR-10), $n = 8$ for ResNet (CIFAR-10) respectively. The CNN-A-Adv, CNN-A-Adv-4, CNN-A-Mix, CNN-A-Mix-4, CNN-B-Adv and CNN-B-Adv-4 models are obtained from the authors or [9] and are the same as the models used in their paper. We summarize the model structures in both incomplete verification and complete verification (Base, Wide and Deep) experiments in Table A1. Our code is available at `http://PaperCode.cc/BetaCROWN`.

### C.2  Additional Experiments

**More results on incomplete verification**    In this paragraph we compare our $\beta$-CROWN FSB to many other incomplete verifiers. WK [42] and CROWN [46] are simple bound propagation methods;

---

[2]Note that our $\beta$-CROWN verifier does not rely on MILP/LP solvers. For these very small MLP models, we find that a MILP solver can actually compute intermediate layer bounds pretty quickly and using these tighter intermediate bounds are quite helpful for $\beta$-CROWN. This also enables us to utilize both CPUs and GPUs on a machine. For all other models, intermediate layer bounds are computed through optimizing Eq. 12. Practically, MILP is not scalable beyond these very small MLP models and these small models are not the main focus of this work.

Table A1: Model structures used in our experiments. For example, Conv(1, 16, 4) stands for a conventional layer with 1 input channel, 16 output channels and a kernel size of $4 \times 4$. Linear(1568, 100) stands for a fully connected layer with 1568 input features and 100 output features. We have ReLU activation functions between two consecutive layers.

| Model name | Model structure |
|---|---|
| CNN-A-Adv (MNIST) | Conv(1, 16, 4) - Conv(16, 32, 4) - Linear(1568, 100) - Linear(100, 10) |
| ConvSmall (MNIST) | Conv(1, 16, 4) - Conv(16, 32, 4) - Linear(800, 100) - Linear(100, 10) |
| ConvBig (MNIST) | Conv(1, 32, 3) - Conv(32, 32, 4) - Conv(32, 64, 3) - Conv(64, 64, 4) - Linear(3136, 512) - Linear(512, 512) - Linear(512, 10) |
| ConvSmall (CIFAR-10) | Conv(3, 16, 4) - Conv(16, 32, 4) - Linear(1152, 100) - Linear(100, 10) |
| ConvBig (CIFAR-10) | Conv(3, 32, 3) - Conv(32, 32, 4) - Conv(32, 64, 3) - Conv(64, 64, 4) - Linear(4096, 512) - Linear(512, 512) - Linear(512, 10) |
| CNN-A-Adv/-4 (CIFAR-10) | Conv(3, 16, 4) - Conv(16, 32, 4) - Linear(2048, 100) - Linear(100, 10) |
| CNN-B-Adv/-4 (CIFAR-10) | Conv(3, 32, 5) - Conv(32, 128, 4) - Linear(8192, 250) - Linear(250, 10) |
| CNN-A-Mix/-4 (CIFAR-10) | Conv(3, 16, 4) - Conv(16, 32, 4) - Linear(2048, 100) - Linear(100, 10) |
| Base (CIFAR-10) | Conv(3, 8, 4) - Conv(8, 16, 4) - Linear(1024, 100) - Linear(100, 10) |
| Wide (CIFAR-10) | Conv(3, 16, 4) - Conv(16, 32, 4) - Linear(2048, 100) - Linear(100, 10) |
| Deep (CIFAR-10) | Conv(3, 8, 4) - Conv(8, 8, 3) - Conv(8, 8, 3) - Conv(8, 8, 4) - Linear(412, 100) - Linear(100, 10) |

CROWN-OPT uses the joint optimization on intermediate layer bounds (optimizing Eq. 12 with no $\hat{\boldsymbol{\beta}}$, as done in [45]). We also include triangle relaxation based LP verifiers with intermediate layer bounds obtained from WK, CROWN and CROWN-OPT. In our experiments in Table 1, we noticed that BIGM+A.SET BABSR [10] and Fast-and-Complete [45] are also very competitive among existing state-of-the-art complete verifiers[3] - they runs fast in many cases with low timeout rates. Therefore, we also evaluate BIGM+A.SET BABSR and Fast-and-Complete with an early stop of 3 minutes for the incomplete verification setting as an extension of Section 4.2. The verified accuracy obtained from each method are reported in Table A2. BIGM+A.SET BABSR and Fast-and-Complete sometimes produce better bounds than SDP-FO, however $\beta$-CROWN FSB consistently outperforms both of them. Additionally, we found that intermediate layer bounds are important for LP verifier on some models, although even with the tightest possible CROWN-OPT bounds the verified accuracy gap between LP verifiers and ours is still large. Additionally, LP verifiers need significantly more time.

**Additional results on the tightness of verification.** In Figure A1, we include LP based verifiers as baselines and compare the lower bound from verification to the upper bound obtained by PGD. The LP verifiers use the triangle relaxations described in Section 2.1, with intermediate layer bounds from WK [42], CROWN [46] and CROWN with joint optimization on intermediate layer bounds (denoted as CROWN-OPT). We find that tighter intermediate layer bounds obtained by CROWN can greatly improve the performance of the LP verifier compared to those using looser ones obtained by Wong and Kolter [42]. Furthermore, using intermediate layer bounds computed by joint optimization can achieve additional improvements. However, our branch and bound with $\beta$-CROWN can significantly outperform these LP verifiers. This shows that BaB is an effective approach for incomplete verification, outperforming the bounds produced by a single LP.

**Lower bound improvements over time** In Figure A2, we plot lower bound values vs. time for $\beta$-CROWN BABSR and BIGM+A.SET BABSR (one of the most competitive methods in Table 1) on the CNN-A-Adv (MNIST) model. Figure A2 shows that branch and bound can indeed quickly improve the lower bound, and our $\beta$-CROWN BABSR is consistently faster than BIGM+A.SET BABSR. In contrast, SDP-FO [9], which typically requires 2 to 3 hours to converge, can only provide very loose bounds during the first 3 minutes of optimization (out of the range on these figures).

**Ablation study of running time on GPUs and CPUs** We conduct the same experiments as in Table 1 but run $\beta$-CROWN FSB on CPUs instead of GPUs. As shown in Table A3, our method is strong even on a single CPU, showing that the good performance does not only come from GPU acceleration; our efficient algorithm also contributes to our success. On the other hand, using GPU can boost the performance by at least 2x. Importantly, the models evaluated in this table are very small ones. Massive parallelization on GPU will lead to more significant acceleration on larger

---

[3]The concurrent work BaDNB (BDD+ FSB) does not have public available code when our paper was submitted.

Table A2: **Verified accuracy (%)** and avg. per-example verification time (s) on 7 models from SDP-FO [9].

| Dataset | MNIST $\epsilon = 0.3$ | | CIFAR $\epsilon = 2/255$ | | | | | | | | | | | |
|---|---|---|---|---|---|---|---|---|---|---|---|---|---|---|
| Model | CNN-A-Adv | | CNN-B-Adv | | CNN-B-Adv4 | | CNN-A-Adv | | CNN-A-Adv4 | | CNN-A-Mix | | CNN-A-Mix4 | |
| Methods | Verified% | Time (s) | Ver.% | Time(s) | Ver.% | Time(s) | Ver.% | Time(s) | Ver.% | Time(s) | Ver.% | Time(s) | Ver.% | Time(s) |
| WK [42] | 0 | 0.1 | 8.5 | 0.4 | 34.5 | 0.8 | 32.5 | 0.4 | 39.5 | 0.5 | 15.0 | 0.3 | 30.0 | 0.4 |
| CROWN [46] | 1.0 | 0.1 | 21.5 | 0.5 | 43.5 | 0.9 | 35.5 | 0.6 | 41.5 | 0.7 | 23.5 | 0.4 | 38.0 | 0.5 |
| CROWN-OPT [45] | 14.0 | 2.7 | 21.5 | 5.5 | 45.0 | 4.0 | 36.0 | 2.0 | 42.0 | 1.6 | 25.0 | 2.3 | 38.5 | 2.0 |
| LP (WK)[§] | 0.5 | 16 | 14.5 | 612 | 41.0 | 1361 | 35.0 | 114 | 41.5 | 140 | 19.0 | 84 | 36.5 | 117 |
| LP (CROWN) | 3.5 | 22 | 21.5 | 941 | 45.0 | 1570 | 36.0 | 123 | 41.5 | 147 | 24.0 | 119 | 38.5 | 126 |
| LP (CROWN-OPT) | 14.0 | 40 | 21.5 | 977 | 45.0 | 1451 | 36.0 | 122 | 42.0 | 152 | 25.0 | 94.8 | 38.5 | 127 |
| SDP-FO [9][*] | 43.4 | >20h | 32.8 | >25h | 46.0 | >25h | 39.6 | >25h | 40.0 | >25h | 39.6 | >25h | 47.8 | >25h |
| PRIMA [26] | 44.5 | 136 | 38.0 | 344 | 53.5 | 44 | 41.5 | 4.8 | 45.0 | 4.9 | 37.5[‡] | 34 | 48.5 | 7.0 |
| BigM+A.Set [10] | 63.0 | 117 | N/A[†] | N/A | N/A | N/A | 41.0 | 79 | **46.0** | 39 | 30.0 | 122 | 47.0 | 71 |
| Fast-and-Complete[§] | 66.0 | 49 | 38.5 | 64 | 51.5 | 21 | 41.5 | 14 | **46.0** | 4.2 | 33.0 | 79 | 46.0 | 10 |
| $\beta$-CROWN FSB | **70.5** | 21 | **46.5** | 32 | **54.0** | 12 | **44.0** | 5.8 | **46.0** | 5.6 | **41.5** | 50 | **50.5** | 5.9 |
| Upper Bound (PGD) | 76.5 | - | 65.0 | - | 63.0 | - | 50.0 | - | 49.5 | - | 53.0 | - | 57.5 | - |

[§] Names in parentheses are methods to compute intermediate layer bounds for the LP verifier.
[*] SDP-FO results are directly from their paper due to the very long running time. All other methods are tested on the same set of 200 examples.
[†] The implementation of BigM+A.Set BaBSR is not compatible with CNN-B-Adv and CNN-B-Adv4 models which have an convolution with asymmetric padding.
[‡] A recent version (Oct 26, 2021) of [26] reported better results on CNN-A-Mix. We found that their results were produced on a selection of 100 data points, and reruning their method using the same command on the same set of 200 random examples as used in other methods in this table produces different results, as reported here.
[§] We use our $\beta$-CROWN code and turn off $\beta$ optimization to emulate the algorithm used in [45]. This in fact leads to better performance than the original approach in [45] because we allow more $\alpha$ variables to be optimized and our implementation is generally better.

Figure A1: Verified lower bound on $f(x)$ by $\beta$-CROWN FSB compared against incomplete LP verifiers using different intermediate layer bounds obtained from [42] (denoted as LP (WK)), CROWN [46] (denoted as LP (CROWN)), and jointly optimized intermediate bounds in Eq. 12 (denoted as LP (CROWN-OPT)), v.s. the adversarial upper bound on $f(x)$ found by PGD. LPs need much longer time to solve than $\beta$-CROWN on CIFAR-10 models (see Table A2).

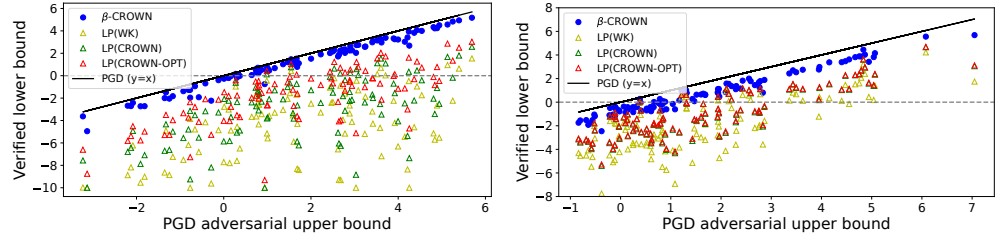

(a) MNIST CNN-A-Adv, runner-up targets, $\epsilon = 0.3$    (b) CIFAR CNN-B-Adv, runner-up targets, $\epsilon = 2/255$

models. The speedup on multi-core CPU is not obvious, possibly due to the limitation of underlying implementations of PyTorch.

**Ablation study on the impact of $\alpha$, $\beta$, and their joint optimization**    We conduct the same experiments as in Table 1 but turn on or turn off $\alpha$ and $\beta$ optimization to see the contribution of each part. As shown in Table A4, optimizing both $\alpha$ and $\beta$ leads to optimal performance. Optimizing beta has a greater impact than optimizing $\alpha$. Joint optimization is helpful for CIFAR10-Base and CIFAR10-Wide models, reducing the overall runtime. For simple models like CIFAR10-Deep,

Figure A2: For the CNN-A-Adv (MNIST) model, we randomly select four examples from the incomplete verification experiment and plot the lower bound v.s. time (in 180 seconds) of $\beta$-CROWN BABSR and BIGM+A.SET BABSR. Larger lower bounds are better. $\beta$-CROWN BaBSR improves bound noticeably faster in all four situations.

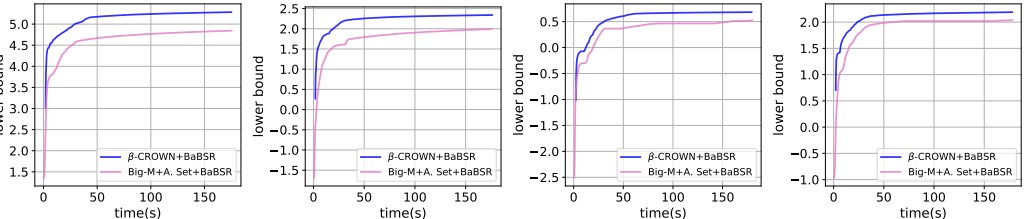

Table A3: Average runtime and average number of branches on three CIFAR-10 models over 100 properties (the same setting as in Table 1) by using different numbers of CPU cores, as well as using a single GPU.

| | CIFAR-10 Base | | | CIFAR-10 Wide | | | CIFAR-10 Deep | | |
|---|---|---|---|---|---|---|---|---|---|
| Hardware | time(s) | branches | %timeout | time(s) | branches | %timeout | time(s) | branches | %timeout |
| 1 CPU | 249.49 | 7886.37 | 4.00 | 178.01 | 2749.96 | 4.00 | 47.46 | 41.12 | 0.00 |
| 4 CPU | 228.28 | 9575.52 | 4.00 | 172.55 | 3956.17 | 4.00 | 45.35 | 41.12 | 0.00 |
| 16 CPU | 222.71 | 10271.08 | 4.00 | 172.40 | 4087.15 | 4.00 | 43.97 | 41.12 | 0.00 |
| 1 GPU | 118.23 | 208018.21 | 3.00 | 78.32 | 116912.57 | 2.00 | 5.69 | 41.12 | 0.00 |

disabling joint optimization can help slightly because this model is very easy to verify (within a few seconds) and using looser bounds reduces verification cost.

Table A4: Ablation study on the CIFAR-10 Base, Wide and Deep models (the same setting as in Table 1), including combinations of optimizing or not optimizing $\alpha$ and/or $\beta$ variables, and using or not using joint optimization for intermediate layer bounds.

| | | | CIFAR-10 Base | | | CIFAR-10 Wide | | | CIFAR-10 Deep | | |
|---|---|---|---|---|---|---|---|---|---|---|---|
| joint opt | $\alpha$ | $\beta$ | time(s) | branches | %timeout | time(s) | branches | %timeout | time(s) | branches | %timeout |
| | ✓ | | 233.86 | 233233.70 | 6.00 | 148.46 | 113017.10 | 4.00 | 5.77 | 260.18 | 0.00 |
| | | ✓ | 174.10 | 163037.05 | 4.00 | 102.65 | 86571.18 | 2.00 | 5.73 | 134.76 | 0.00 |
| | ✓ | ✓ | 139.83 | 133346.44 | 3.00 | 91.01 | 73713.30 | 2.00 | 5.22 | 100.44 | 0.00 |
| ✓ | ✓ | | 163.69 | 160058.80 | 4.00 | 149.00 | 115509.71 | 4.00 | 8.58 | 65.70 | 0.00 |
| ✓ | | ✓ | 162.95 | 150631.49 | 4.00 | 89.22 | 72479.96 | 2.00 | 8.38 | 52.26 | 0.00 |
| ✓ | ✓ | ✓ | 118.23 | 208018.21 | 3.00 | 78.32 | 116912.57 | 2.00 | 5.69 | 41.12 | 0.00 |