# OpenReview forum: "Beta-CROWN: Efficient Bound Propagation with Per-neuron Split Constraints for Neural Network Robustness Verification"
_NeurIPS.cc/2021/Conference — NeurIPS 2021 Poster_

### Official Review · Reviewer_pv7s · 2021-07-09

**Rating:** 6
**Confidence:** 5

**Summary:**

The paper presents Beta-CROWN, a novel algorithm to compute bounds on the output of neural network activations.
Beta-CROWN is designed to be employed within recent branch-and-bound frameworks [6, 10], resulting in an effective complete (or incomplete, when returning early) verification algorithm.

The authors build on the bounding algorithm presented in [43] ("alpha-CROWN"), which (with fixed intermediate bounds) is an effective solver for the popular triangle LP relaxation (Planet [14]) of ReLU neural networks.
The main contributions of the paper are the following:
- differently from alpha-CROWN, beta-CROWN can represent the effect that a ReLU split at the i-th layer has on the activations preceding it, leading to tighter bounds within branch and bound;
- when jointly optimizing over intermediate bounds and output bounds, beta-CROWN shares fewer parameters between the various sub-problems, resulting in tighter final bounds.

**Limitations And Societal Impact:**

I could not find any detailed description of the limitations of the paper. While most of the limitations are shared with many works in the area of neural network verification, the authors could point out to the fact that proposed method only deals with ReLU activations, or state the assumptions on the input-output specifications of the networks to be verified (for instance, convexity of the input set).

**Main Review:**

**Strengths:**
Alpha-CROWN [43] has shown the potential of optimized propagation-based methods within branch and bound for neural network verification, resulting in very fast complete verification on easier verification properties.
Beta-CROWN successfully addresses the main weakness of alpha-CROWN (lack of split constraint support), enhancing performance on the harder properties and improving upon the considered baselines.

**Concerns:**
While the empirical results look good, I believe the presentation of the paper is rather confusing. Beta-CROWN is an incremental improvement over alpha-CROWN, the main difference being the addition of split constraints in the formulation (see equation (32) for a dual perspective).
While this is not a problem per se, the connection seems to be systematically understated in the paper: for instance, alpha-CROWN should be properly described in section 2, whereas the presentation in section 3 rarely mentions alpha-CROWN [43] and does not cite it when introducing the optimization over the alpha variables (lines 197-200). As made clearer in the Appendix, the results in section 3.2 follow from results in alpha-CROWN with only minor modifications to include split constraints (yet this discussion is absent in the main paper).
Furthermore, from a quick reading, it would appear like beta-CROWN is the first GPU-accelerated dual bounding algorithm (excluding LP solvers) that can incorporate split-constraints (lines 136,137,149), however they were naturally incorporated by some previous methods such as [5, 9, 12] (as mentioned by the authors in Appendix B.2).

I feel like ablation studies are missing: the importance of the joint optimization of intermediate bounds is often stressed. However, how much improvement is due to the split constraints, and how much to the joint optimization? What would be the performance if tight intermediate bounds were computed only once, before any branching? In addition, it would be great if the authors could investigate the performance of the algorithm if the alpha variables from alpha-CROWN are not optimized over.

The authors could perhaps tone down the claims on the relative performance of Beta-CROWN to other GPU-based verifiers (lines 59-60). For instance, Beta-CROWN on average does not appear to be "2 to 7" times faster than [10] (and [9]) on the Wide and Deep networks of Table 1. Moreover, the relative improvement over [10] further decreases when part of its contributions (the FSB branching strategy) are not integrated within Beta-CROWN.

Finally, the incomplete verification results (verified accuracy) in the main paper should include more branch-and-bound-based incomplete verifiers such as alpha-CROWN [43] (and the experiments on [9] presented in the Appendix), which may be stronger baselines than the ones considered.

**Time Spent Reviewing:**

7

---

> ### Author Response · Authors · 2021-08-11
> **Thank you for your constructive feedback, we will revise paper accordingly and we provided additional results.**
>
> We thank the reviewer for the constructive comments and we will fix the presentation issues mentioned in your review, and we have conducted the requested ablation study experiments. Additionally, we will also add a paragraph discussing limitations and societal impact. We provide detailed answers to your questions below:
>
> **Q1. Discussions on alpha-CROWN [43]**
>
> We thank the reviewer for the valuable comments and suggestions on paper presentations.  In the next version of our paper, we will introduce alpha-CROWN in Section 2.2, emphasize the connection early in Section 3 (line 192-200), make sure to cite [43] when introducing the alpha formulation, and describe previously proposed GPU-based complete verifiers in Section 1 (it is currently in Appendix B.2).
>
> We emphasize that our handling of split constraints is *not an incremental improvement* over alpha-CROWN. First, alpha-CROWN itself cannot be used to build a complete verifier, and [43] still relies on a linear programming (LP) verifier on CPU to achieve completeness; we achieve completeness (Theorem 3.3) and are capable of full GPU acceleration. Second, when empirically compared to alpha-CROWN (Fast-and-Complete [43]), we have achieved much lower timeout rates (17% vs. 3% on CIFAR-10 Base, 9 vs. 2% on CIFAR-10 Wide in Table 1) and also faster mean verification time; the improvements are not incremental. Furthermore, as stated in Section 3.3 (line 234-240) and Appendix B.2, the underlying formulation for joint optimization with $\alpha$ is also different from [43] as our formulation will have a much larger searching space for the $\alpha$ variables, leading to even tighter bounds.
>
> **Q2: Additional ablation studies**
>
> We presented additional ablation studies based on the reviewer’s suggestions in [Table R2](https://beta-crown.github.io/#table-r2-ablation-study-of-alpha-beta-and-joint-optimization).
> . We conduct a few ablation studies on the CIFAR-10 base, wide and deep models (same as those in Table 1), with different options of optimizing only $\alpha$ or $\beta$, and with and without joint optimization of bounds. In all of our experiments, we found that it is always helpful to compute tight intermediate bounds using joint optimization of intermediate layers before branching, and optimize only the output layer’s $\alpha$ and $\beta$ during branching as it is usually much faster. We will make this point clear in our paper.
>
> Table R2 - Ablation Study of joint bound optimization, and optimization over only $\alpha$ or only $\beta$: https://beta-crown.github.io/#table-r2-ablation-study-of-alpha-beta-and-joint-optimization
>
> 1. How much improvement is due to the split constraints (beta optimization)?
>
> We disabled beta optimization and found that the results became much worse. The results can be found in the 1st and 4th row of [Table R2](https://beta-crown.github.io/#table-r2-ablation-study-of-alpha-beta-and-joint-optimization) where beta optimization is not enabled. For example, without using beta optimization, the time increases from 139.83s to 233.86s on the Base model when joint optimization is disabled. Same tendency can be found in all other settings and models. Also the timeout rates are the worst. Disabling beta optimization has a bigger negative impact compared to disabling $\alpha$ optimization and optimizing only $\beta$. Thus, it is important to consider split constraints.
>
> 2. How much improvement is due to the alpha optimization?
>
> We disabled $\alpha$ optimization and kept only beta optimization. The results can be found in row 2 and 5 of [Table R2](https://beta-crown.github.io/#table-r2-ablation-study-of-alpha-beta-and-joint-optimization) where $\alpha$ optimization is not enabled. Generally, it performs worse than optimizing both $\alpha$ and $\beta$, but performs better than optimizing $\alpha$ only. For example, without using $\alpha$ optimization, the time increased from 139.83s to 174.10s on the Base model when joint optimization is disabled, and increased from 122.71s to 162.95s  when joint optimization is enabled. Same tendency can be found in all other models.
>
>
> 3. How much improvement is due to the joint optimization of intermediate bounds?
>
> As shown in [Table R2](https://beta-crown.github.io/#table-r2-ablation-study-of-alpha-beta-and-joint-optimization), joint optimization is helpful for CIFAR10-Base and CIFAR10-Wide models, reducing the overall runtime. For simple models like CIFAR10-Deep, disabling joint optimization can help slightly because this model is very easy to verify and using looser bounds reduces verification cost.
>
>
> **Q3. Relative Performance Improvement, and branching strategy**
>
> As you suggested, we will remove the 2-7 time speedup claim. The claim was added to our paper before we added a concurrent preprint [10] (on Arxiv in April) into comparison. However, our method still clearly outperforms all baselines in both verification time and timeout rates.
>
> We include the FSB branching in evaluation for a fair comparison to [10], because [10] is essentially BDD+ [5] with FSB. We report Beta-CROWN with and without FSB. Our approach can benefit from any improvements on branching heuristics, although even without the new FSB heuristic, we are still the fastest in Table 1, even when compared to the baselines with FSB.
>
>
> **Q4: More experiment results**
>
> In Table 5, in appendix, we have listed 10 verifiers for incomplete verification, including the strongest one, Big.M + A.set in Table 1 (BaDNB is better but we contacted the authors and found out that its source code was not available). Fast-and-Complete tends to perform worse than Big.M + A.set in Table 1 so we did not include it originally. In [Table R3](https://beta-crown.github.io/#table-r3-baseline-results-for-fast-and-complete), we have now included the numbers for Fast-and-Complete [43] on these models. It generally performs worse than Beta-CROWN in all settings.
>
> Table R3 - Baseline results for Fast-and-Complete [43]: https://beta-crown.github.io/#table-r3-baseline-results-for-fast-and-complete
>
> In the [general response](https://openreview.net/forum?id=ahYIlRBeCFw&noteId=jjB71Dx5jwQ), we also provide updated and improved results in [Table 1](https://beta-crown.github.io/#updated-table1), [Figure 1](https://beta-crown.github.io/#updated-figure1), [Table 2](https://beta-crown.github.io/#updated-table2), and [Table 3](https://beta-crown.github.io/#updated-table3), with a few additional models (MLP models from ERAN) and recent code improvements. For the CIFAR-10 Base, Wide and Deep models we are able to verify 80% instances under 10 seconds, where existing verifiers can take much longer time especially on the CIFAR-10 Base model. We hope the reviewer can take a look at these new results as well.
>
> **Q5: Limitations**
>
> In [general response](https://openreview.net/forum?id=ahYIlRBeCFw&noteId=jjB71Dx5jwQ), we provided a paragraph on limitations and future works that we will add to the next version of the paper. Specifically, non-ReLU-based NNs and more general input-output verification specifications remain to be the limitations of our paper and many other concurrent works in NN verification, and could potentially be very interesting future directions. Let us know if you have any further comments or suggestions on this part.
>
>
> We would like to thank the reviewer again for the constructive feedback. We have conducted all requested ablation studies and required new experiments, and we hope the reviewer can reevaluate our paper based on the new empirical results and our response. Thank you.

---

### Official Review · Reviewer_kL3f · 2021-07-13

**Rating:** 7
**Confidence:** 4

**Summary:**

The authors propose an augmentation to CROWN that leverages weak duality to enable neuron-split constraints. This approach is amenable to GPU acceleration and can theoretically attain (or even surpass) the bounds produced by the costlier LP based incomplete verification methods.

**Limitations And Societal Impact:**

The limitations of this approach were not addressed, nor were the societal impacts. For methods guaranteeing safety of neural networks, the societal impacts are mostly beneficial and can be read in many other papers. It would be nice to see the authors address the potential limitations of this work.

**Main Review:**

Originality:
At its heart, this approach is a small tweak to well-established techniques that introduces a new set of optimization variables (the beta variables) than can then be optimized in a similar fashion to the alpha's (i.e. in the Fast-and-complete paper). The notation is heavy, but the idea itself is quite simple. Similarly, the proofs are only minor modifications to existing results. The related work is clearly and fairly compared against, and it the exact contributions of this work are clearly delineated.

Clarity:
The paper is clearly written and easy to follow (for CROWN-style papers, which are notoriously heavy in their notation). The lengthy derivations are conveniently pushed to the appendices and only the relevant details are kept in the main paper. In some of the experiments, it is not

Quality:
The experimentation section is very thorough and extremely convincing. The approach presented by the authors is the new state-of-the-art for both complete and incomplete neural network verification on a standard set of benchmark networks and properties. I have very little to complain about here: while the tweak to CROWN is fairly minor, the resulting method wins in both the verification% and time against the extensive set of benchmarks considered. Certainly it would be interesting to also consider cases outside of the VNN suite of metrics, such as i) ablating against a lack of GPU to demonstrate how much GPU acceleration helps here, ii) considering different domains outside of image classification or the standard benchmark models, iii) considering different threat models; but the lack of these results does not detract from the paper. From a theoretical standpoint, the comparisons to the dual form and arguments that Beta-CROWN can attain the same LP bounds with the same intermediate layer bounds (and improve when the intermediate layer bounds are adaptive), is a nice addition to the paper and increases the strength of this submission.

Significance:
The problem of verifying neural networks against adversarial inputs is very important, and this work extends SOTA in that domain. Bound propagation methods, or complete verification techniques, however, still do not show the promise of being able to provide meaningful guarantees against large SOTA image classifiers (such as Resnet50, etc). While this work certainly is better than existing works, I'm hesitant to highly praise a line of work that seems unlikely to ever be applicable to real-world networks, or even the standard threat-models for the small networks. Chipping away at faster and tighter bounds on small networks and small threat models seems like a great way to get papers published, but I require more convincing that this line of work has the potential to be applicable to the networks or threat models used in practice.

Overall:
This paper clearly explains an approach that modifies existing techniques to extend the state of the art for a problem that is important and receives significant attention within the community. I am dubious of this approach for long-term success or impact, but for the stated problem, this is the best solution we have right now and I'd like to see this paper accepted.

Some typos:
line 108 and 506: inequalities should be flipped to show $l <= v <= u$


**Time Spent Reviewing:**

4

---

> ### Author Response · Authors · 2021-08-11
> **Thank you for the very insightful comments and we address your questions below.**
>
> Thank the reviewer for recognizing our contributions and valuable comments and suggestions. We will address each of your questions and concerns below:
>
> **Q1: Originality**
>
> We agree with the reviewer that our main idea is simple, but this simple idea works quite well, outperforming many more complicated approaches and has achieved great impact on the field of VNN. We believe the simplicity of our approach is a plus, as it allows very efficient implementation and easier adoption. Although handling the split constraint via the $\beta$ constraints looks straightforward, it is crucial to our success: compared to Fast-and-Complete which optimizes only the $\alpha$ constraints, we have much lower timeout rates (17% vs. 3% on CIFAR-10 Base, 9 vs. 2% on CIFAR-10 Wide in Table 1).
>
> **Q2: Clarity**
>
> We thank the reviewer for the positive comments on clarity. It seems several sentences in your review were missing in the “clarity” paragraph. Please kindly let us know if the missing sentences contain any questions or concerns and we will be more than happy to address them.
>
> **Q3: Quality, additional experimental results**
>
> We are very glad to know that our experimentation section is thorough and extremely convincing. As suggested, we present the additional ablation studies on CPUs and GPUs. The running time of our method on the CIFAR-10 Base, Wide, and Deep models are shown in [Table R1](https://beta-crown.github.io/#table-r1-ablation-study-of-running-time-on-gpus-and-cpus). When using 1, 4, and even 16 CPU cores without a GPU, the average verification time is 2X or slower than using a single GPU. We also found that using additional CPU cores did not significantly improve performance, possibly due to the limitation of PyTorch (we did set the number of threads correctly and observed that multiple CPU cores were utilized, but the speedup is very poor). We also note that these models are very small, and the benefits of GPU acceleration will be more obvious on larger models. For example, for the ResNet and ConvBig models in Table 2, optimizing the bounds for 1 iteration can take a few seconds on CPUs, and only less than 100 milliseconds on a GPU.
>
> Table R1 - Ablation study of running time on GPUs and CPUs: https://beta-crown.github.io/#table-r1-ablation-study-of-running-time-on-gpus-and-cpus
>
> Additionally, we provided new experimental results in the [general response](https://openreview.net/forum?id=ahYIlRBeCFw&noteId=jjB71Dx5jwQ), where more models are tested and our implementation is further improved, yielding better results. We hope the reviewer can take a look at these new results as well.
>
> Thank you for suggesting considering problems in different domains and different threat models. They are great suggestions for advancing the field of VNN. In this paper, we focused on standardized benchmarks but we aim to extend our approach to additional domains under more general threat models in future work.
>
> **Q4: Significance and scalability of complete verification**
>
> While we totally agree that no approach to-date is strong enough to provide formal verification for large classifiers and datasets like ResNet50 and ImageNet, the field of neural network verification is still young and Rome was not built within a day.
>
> In the past 5 years, the VNN community has achieved significant improvements. In 2017, the first verifier Reluplex [19] could take hours to verify an MLP model with 5 inputs and 300 hidden nodes. In 2018, branch and bound based complete verification were proposed [6], which can solve networks with a few thousands neurons within an hour. Very recently, GPU accelerated verifiers, such as Beta-CROWN, scale up the capability of complete NN verification to even larger networks while greatly reducing verification time. Compared to the basic BaB based complete verifier, BaBSR [6] (2018), in figure 1 we have demonstrated three orders of magnitude speedups. Possibly in the next 5 years, future complete verifiers can scale to large networks like ResNet50 with more advanced techniques. We do believe our work is an important milestone in the quest for scalable neural network verifiers.
>
> **Q5: Limitations and Societal Impacts**
>
> In our [general response](https://openreview.net/forum?id=ahYIlRBeCFw&noteId=jjB71Dx5jwQ), we provided a paragraph on limitations, future works and the societal impacts of our work, which we will add to the next version of the paper. Specifically, we will investigate BaB for non-ReLU-based NNs, and discuss problems outside of image classifications and with different realistic threat models. These challenges remain to be the limitations of many concurrent works in NN verification, and could potentially be very interesting future work directions. Additionally, our algorithm can be used to guarantee the safety and robustness of neural networks and has overall positive societal impacts (detailed in the [general response](https://openreview.net/forum?id=ahYIlRBeCFw&noteId=jjB71Dx5jwQ)). Let us know if you have any comments or suggestions on our discussions about limitations and societal impacts.
>
>
> We thank the reviewer again for the very insightful comments, and we hope that our response has addressed your concerns on our paper.

---

### Official Review · Reviewer_vGgK · 2021-07-16

**Rating:** 7
**Confidence:** 4

**Summary:**

The paper develops a bound propagation method for neural network verification.
Differently from related work, the method adds support for ReLU split constraints
thereby offering a more efficient way (than linear programming) to bound a
verification problem under the constraints.


**Limitations And Societal Impact:**

The authors adequately addressed the limitations of their work. There is no discussion on potential negative societal impact of their work.

**Main Review:**

Bound propagation methods were shown particularly effective in both incomplete
and complete verification, where in the latter case they are commonly used to
derive strong MILP formulations of the verification problem in question.
However, in the case of customised BaB methods for neural network verification,
where the verification sub-problems resulting from branching on the ReLU nodes
need to be bounded, propagation methods cannot handle the ReLU constraints
introduced from branching. I believe that the paper makes a significant
contribution towards overcoming this shortcoming, thus contributing to improved
scalability in complete verification.  Indeed, the extensive experimental
results speak for themselves of the attractiveness of the approach.

The paper is very well written and easy to follow. Some typos that I found:

Line 108: u \leq \upsilon \leq l -> l \leq \upsilon \leq u ?
Line 135: intersection -> union ?



**Time Spent Reviewing:**

3

---

> ### Author Response · Authors · 2021-08-11
> **Thank you for the positive and encouraging review!**
>
> We greatly appreciate the reviewer for recognizing the importance of our work, accurately summarizing our main contributions and giving encouraging comments. Thank you for identifying the typos in our submission and we will fix them in our final revision. Additionally, we provided new experimental results in the [general response](https://openreview.net/forum?id=ahYIlRBeCFw&noteId=jjB71Dx5jwQ), where more models are tested and our implementation is further improved, yielding better results. Please kindly let us know if you have any additional comments or suggestions.

---

### Official Review · Reviewer_qX8M · 2021-07-24

**Rating:** 6
**Confidence:** 4

**Summary:**

The paper deals with efficiently incorporating CROWN into the branch-and-bound framework for complete verification. The authors propose to relax the split constraints using Lagrangian relaxation, and show how this can be done within the bound propagation framework. They also jointly optimise the intermediate bounds to tighter verification.

The technical contribution is fairly simple.

The method gives good improvements over existing complete verification methods. Authors have done a very good experimental evaluation, on different datasets and compared with many baselines.

The writing is good overall. Sections 2 and 3 are very well written. I have some comments which I have added in the main review.

The related work section and conclusion section need some work.


**Limitations And Societal Impact:**

The paper does not discuss limitations or societal impact. A potential negative societal impact is that since verification exposes potential flaws in networks, it can be misused too.

**Main Review:**

The paper proposes a new method for neural network verification (mainly complete verification), which gives state-of-the-art results on a wide range of datasets. The authors propose a technique to deal with neuron split constraints while using CROWN within branch-and-bound.

The key idea is to use Lagrangian relaxation to relax the split constraints. This is done in a way such that linear bound propagation can still be used. The idea is fairly simple, and the method section is well written. The authors also jointly optimize the intermediate bounds.

This technique is very useful for the verification community. CROWN has been a state-of-the-art incomplete verifier but has not had much success in complete verification. This technique allows the use of efficient CROWN style algorithms to speed up complete verification.

I have a few comments:
1. Introduction
- I don't see why the authors have not added the details about using Lagrangian relaxation to relax the split constraints in L46-48? It would help better understand what alpha and beta are in L48.

2. Section 3
- The layer constraints seem to be missing from Eq 2.
- L186-188 are good. This intuition can be brought forward to the introduction too. It takes 4 pages before anything about the actual method is brought. Giving detail about the Lagrangian function in the introduction would help better capture the interest of the reader.

3. Experimental Results
- In section 4.1, are all these methods run on the same machine (GPU and CPU)?
- The colours in Fig.1 are quite confusing. It takes many minutes to understand the figure, maybe remove some baselines to avoid confusion.

4. Related Work
- L334 discussion about [35] is factually incorrect. They do not relax a layer of neurons. They give a relaxation for composition of linear+ReLU. Please be careful when writing related work.
- L336 I don't think you are breaking the convex barrier since you are still using the planet relaxation. From your definition, all bab methods will break the barrier. I don't think that is the point that the convex barrier paper is trying to make.
- The main comparing baselines seem to be [9, 10 and 42]. So discuss these more thoroughly. The description about [9] is not clear.

6. Conclusion
- No future work is mentioned.
- No limitations mentioned.
- No broader impact discussed.
Please add these


====================================

***Updates after rebuttal***

I would like to thank the authors for answering the questions. I hope that the authors will indeed add the limitations in the main paper. My initial questions were sufficiently answered by the authors.
However, some points raised by other reviewers are valid. I also believe that the technical delta over alpha-crown is incremental.
The experimental results are good and I am still voting for acceptance.


**Time Spent Reviewing:**

5

---

> ### Author Response · Authors · 2021-08-11
> **Thank you for the very helpful reviews!**
>
> We thank the reviewer for recognizing the importance of our work and the valuable suggestions for improving the presentations of the paper. We really appreciate the constructive comments and answer specific questions below:
>
> **Q1, Q2: Introduction and Section 3**
>
> We will highlight the intuition of Lagrangian relaxation for efficiently solving neuron split constraints in the introduction. Thank you for the very good suggestion. We will also fix the missing constraint in Eq. 2.
>
> **Q3: Experiments**
>
> We apologize for too many colors in the figure, and we will move some baselines to the appendix for clarity. Beta-CROWN (BaBSR), Beta-CROWN (FSB) and Fast-and-Complete were run on the same machine with one CPU and an Nvidia 1080 Ti GPU. Other baseline results were mostly from results published in earlier works. Results for our main competitors, A. set BaBSR and BigM+A. set BaBSR were produced on an Nvidia Titan Xp GPU [9], which is actually better hardware than ours.
>
> Additionally, we provided new experimental results in the [general response](https://openreview.net/forum?id=ahYIlRBeCFw&noteId=jjB71Dx5jwQ), where more models are tested and our implementation is further improved, yielding better results. We hope the reviewer can take a look at these new results as well.
>
> **Q4: Related Work**
>
> We will fix the discussion about [35], which relaxes a linear layer with a ReLU neuron. We will also add more discussions for [9] in the main text (currently the discussions are in appendix B.2).
>
> We agree with the reviewer that we still rely on Planet relaxation for unstable ReLU neurons not split by BaB and are subject to the convex relaxation barrier for each subproblem, although the convex relaxation barrier paper does list BaB as a way to break the barrier (see Appendix A “How to break the barrier” in [29]), in the sense that BaB eliminates unstable ReLU neurons and give tighter bounds compared to a single Planet relaxation. To avoid potential confusion, we will make it clear that each subproblem in BaB is still subject to the barrier. Feel free to let us know if you have further comments on this.
>
> **Q5, Q6: Conclusions, Limitations, Impacts**
> We will add discussions on future work including using split bounds for certifiable defense, studying tighter bound propagations and improving the scalability to more difficult verification instances, and performing efficient BaB based verification for non-ReLU NNs. We will also discuss limitations and broader societal impacts as elaborated in the [general response](https://openreview.net/forum?id=ahYIlRBeCFw&noteId=jjB71Dx5jwQ).
>
> In the [general response](https://openreview.net/forum?id=ahYIlRBeCFw&noteId=jjB71Dx5jwQ), we provided detailed changes for improving the presentation, addressing all the suggestions and comments from the reviewers. Please kindly let us know if you have any additional comments or find anything unsatisfying. Thank you very much!

---

> ### Author Response · Authors · 2021-08-31
> **We are confused to see our paper rating decreased without new comments from the reviewer**
>
> Dear Reviewer qX8M,
>
> We thank you again for your constructive feedback. We just noticed that you decreased the rating for our paper but did not leave an updated review. We are unsure why the rating was decreased after our response. If there is anything that you find confusing or unsatisfying in our response, please kindly let us know.
>
> In our response below, we detailed the changes to enhance the presentation of our paper according to your suggestions, and in the general response, we also detailed the limitations and societal impact section we are going to add to our paper. Additionally, we added extensive ablation studies, and further improved our code. Our updated results show even more performance gains compared to SOTA baselines. For example, on the CIFAR-10 Base benchmark we can verify 80% instances **under 10 seconds** and all existing approaches need **over a minute** to do so. All new results added during the discussion period can be found on an anonymous website (https://beta-crown.github.io/). We do believe our paper makes a substantial contribution to the field of NN verification.
>
> We are confused to see that the paper rating has decreased after our additional effort during the discussion period. We hope the reviewers can communicate with us and let us know if there is anything else we need to address before the discussion period is closed and reconsider the evaluation. Thank you.
>
> Sincerely,
> Anonymous Authors

---

> ### Author Response · Authors · 2021-09-02
> **Thank you for your support! We want to point out again our main contributions.**
>
> Dear Reviewer qX8M,
>
> We thank you very much for updating your review and voting for accepting our paper. We really appreciate your support. We will make sure to add the paragraph discussing limitations to our paper.
>
> We do want to respectfully point out that **our technical contributions are not incremental**. The main connection between our work and $\alpha$-CROWN is that we also optimize the slopes variable $\alpha$. This is never the main point of our paper and is only briefly discussed. The main technical contribution of our paper includes:
>
> 1. We are the **first bound propagation based method that can handle split constraints**. In $\alpha$-CROWN, split constraints cannot be enforced, leading to looser bounds. Previously, split constraints were often handled by expensive algorithms such as linear programming (LP) which are slow and not GPU-accelerable.
> 2. Our $\beta$ variables were from Lagrangians, but $\alpha$ are not - they are the lower bound slopes of ReLU neurons. So our variables are developed in a fundamentally different way.
> 3. Compared to $\alpha$-CROWN, we achieve **complete verification purely on GPUs** without the assistance of LP solvers. In the $\alpha$-CROWN paper, they had to use LP solver to guarantee the completeness, which cannot be fully GPU accelerated [43]. LP in $\alpha$-CROWN is necessary, otherwise complete verification cannot be achieved (see discussion under their Theorem 3.2).
> 4. Our **newly added ablation study** shows that optimizing $\beta$ is more important than optimizing $\alpha$ on the CIFAR-10 benchmarks. In [Table R2](https://beta-crown.github.io/#table-r2-ablation-study-of-alpha-beta-and-joint-optimization), you can find that just optimizing $\beta$ is more effective (shorter avg. runtime) than just optimizing $\alpha$ on three models, no matter whether the intermediate bounds are jointly optimized (“joint opt”). In [Table R3](https://beta-crown.github.io/#table-r3-baseline-results-for-fast-and-complete) we also added $\alpha$-CROWN (Fast-and-Complete) on 7 more models and you can see the performance gap is large on many models.
>
> Our paper was written in an easy-to-understand way that tends to closely follow the formulations in existing works, which might make it look incremental on the surface. Although we could have written it in a more cryptic way so the connections to prior works are harder to see, we do not feel that is the right way. We hope the reviewer can understand our main contributions better now and reevaluate our paper based on our response. Many thanks.
>
> Sincerely,
> Anonymous Authors

---

### Author Response · Authors · 2021-08-11
**General Response: Experimental results further Improved and new models added. We will also add a discussion on limitations and societal impacts.**

**New Experimental Results (better performance, more models compared)**

We provide an updated version for the main results tables and figures. The code submission used in our original submission was preliminary and not well optimized, and we have improved the implementation during the past few weeks. This allows us to demonstrate even stronger results - please see the updated tables and figures below. For all three CIFAR-10 Base, Wide and Deep models in [updated Figure 1](https://beta-crown.github.io/#updated-figure1), we can solve over 80% of instances within just 10 seconds, and many existing methods take 10X longer time on the Base model. In  [updated Table 2](https://beta-crown.github.io/#updated-table2), the verified accuracy is further improved, and verification time is reduced.

Additionally, in the [updated Table 2](https://beta-crown.github.io/#updated-table2), we added the MLP models used in benchmarks of [33][35][26], for a more comprehensive evaluation. We strongly outperform existing methods. We will add discussions of these models in our final revision and add experiment details in the appendix.

Updated Table 1: https://beta-crown.github.io/#updated-table1

Updated Figure 1: https://beta-crown.github.io/#updated-figure1

Updated Table 2: https://beta-crown.github.io/#updated-table2

Updated Table 3: https://beta-crown.github.io/#updated-table3



**Limitations of Beta-CROWN**

We thank the reviewers for pointing out some limitations of our approach, and most of these limitations also apply to most NN verifiers and are good directions for future research. We will add the following paragraph in the final version of the paper discussing the limitations of our approach and potential future work directions:

Our $\beta$-CROWN BaB verifier has several limitations which are commonly shared with most existing BaB-based complete verifiers. First, we focused on ReLU activation functions which can be split into two linear cases. For other non-piecewise linear activation functions, although it is still possible to conduct branch and bound, it is difficult to guarantee completeness because the activation function is still nonlinear and requires relaxation via CROWN. Second, we discussed only the $\ell_p$ norm perturbations for input domains in Section 3, and most verifiers including ours focused on $\ell_\infty$ norm perturbation in experiments. In practice, the threat model may involve complicated and nonconvex perturbation specifications, which cannot be directly handled by Beta-CROWN. Third, although our GPU accelerated verifier outperforms existing ones, all BaB based verifiers, including ours, are still limited to relatively small models far from the ImageNet scale. Finally, we have only demonstrated Beta-CROWN BaB for verifying the robustness of image classification tasks, and generalizing our methods to give verification guarantees for other tasks such as reinforcement learning is an interesting direction for future work.

**Societal Impacts**

We will add the following paragraph in the final version of our paper to discuss societal impacts of this work.

Neural networks (NNs) have been used in an increasingly wide range of real-world applications and play an important role in artificial intelligence (AI). The trustworthiness and robustness of NNs have become crucial factors since AI plays an important role in modern society. Beta-CROWN is a strong neural network verifier which can be used to check certain properties of neural networks, which can be helpful for guaranteeing the robustness, correctness, and fairness of NNs in applications that can directly or indirectly impact human life. We believe our work has overall positive societal impacts, although it may potentially be misused to identify the weakness of NNs and guide attacks.

---

### Author Response · Authors · 2021-08-27
**We thank the reviewers again and please let us know if you have any further questions before the discussion is closed**

Dear Reviewers,

We really appreciate your constructive and encouraging comments. As suggested by the reviewers, in our response we included **comprehensive ablation study experiments**, and we will discuss limitations and societal impacts of our paper and further improve clarity according to reviewers’ suggestions. We also answered each reviewer’s questions in detail. Additionally, we further improved our code and our updated results show **even more performance gains** compared to baselines. All new results can be found on an anonymous website: https://beta-crown.github.io/

Because the discussion period is ending soon, we hope the reviewers can take a look at our response, and please kindly let us know if there are any further questions or concerns. Thank you.

Sincerely,
Anonymous Authors

---

### Decision · Program_Chairs · 2021-09-27

**Decision:**

Accept (Poster)

**Comment:**

The paper was well-received. The main idea is fairly simple, but the problem is important and the writing and empirical evaluation are solid. Based on the reviewers' advice, I am recommending acceptance. Please make sure to add to the final version the new results and the text on limitations and societal impact that you described in your author response.